# The Impact of Achievements in Mathematics on Cognitive Ability in Primary School

**DOI:** 10.3390/brainsci12060736

**Published:** 2022-06-03

**Authors:** Irina Kliziene, Asta Paskovske, Ginas Cizauskas, Aldona Augustiniene, Berita Simonaitiene, Ramunas Kubiliunas

**Affiliations:** 1Educational Research Group, Institute of Social Science and Humanity, Kaunas University of Technology, 44249 Kaunas, Lithuania; aldona.augustiniene@ktu.lt (A.A.); berita.simonaitiene@ktu.lt (B.S.); 2Study Programme “Information Technologies of Distance Education”, Faculty of Informatics, Kaunas University of Technology, 44249 Kaunas, Lithuania; asta.paskovske@ktu.edu; 3Department of Mechanical Engineering, Faculty of Mechanical Engineering and Design, Kaunas University of Technology, 44249 Kaunas, Lithuania; ginas.cizauskas@ktu.lt; 4Department of Software Engineering, Faculty of Informatics, Kaunas University of Technology, 44249 Kaunas, Lithuania; ramunas.kubiliunas@ktu.lt

**Keywords:** primary education, cognitive ability, mathematics achievement, problem solving

## Abstract

Cognitive skills predict academic performance, so schools that try to improve academic performance might also improve cognitive skills. The purpose of this study was to determine the effect of achievements in mathematics on cognitive ability in primary school. Methods: Participants: 100 girls and 102 boys aged 9–10 years (the fourth grade) were selected from three schools. A *diagnostic test of cognitive abilities* (DTCA) was created by the authors of the article for the assessment of primary school students’ cognitive abilities. The diagnostic cognitive ability test was based on Reuven Feuerstein’s theory of dynamic cognitive modality assessment, the problem-solving model, and followed the mathematics curriculum for grade 4. The tasks of the test were distributed according to the cognitive function: systematic exploration, spatial orientation, sequencing, image recognition, recognizing and understanding relationships, collecting and processing information, algorithm development, data management (classification), and construction of combinations. *Achievements in mathematics:* they were collected systematically using short- and medium-term mathematics tests, and the levels of achaievement were defined of grade 4 primary school students to assess individual learner performance, anticipate their learning strengths and weaknesses, and shape their subsequent learning process. Results: With regard to the relationships between cognitive functions and achievement level, Spearman’s correlation analysis revealed the relationships between the following cognitive functions: systematic exploration and spatial orientation (Spearman q = 0.276, *p* = 0.022), systematic exploration and designing an algorithm development (Spearman q = 0.351, *p* = 0.003), spatial orientation and data management (Spearman q = 0.274, *p* = 0.023), sequencing and combination construction (Spearman q = 0.275, *p* = 0.022), and sequencing and recognizing and understanding relationships (Spearman q = 0.243, *p* = 0.044). Conclusions: (1) The internal validity of the diagnostic test of cognitive abilities was supported by significant correlations between cognitive functions and mathematics achievement. This suggests that this methodology of the diagnostic cognitive ability test can be used to assess the cognitive abilities of primary school students. (2) The diagnostic test of cognitive abilities showed that the majority of primary school students reached higher levels of achievement in a systematic inquiry (systematic, non-impulsive, planned behavior when collecting data or checking information). A difference was observed in the ability of students to navigate in space and follow directions for primary school students at a satisfactory or higher level. Primary school students’ performance in identifying the rule for the sequencing of elements, finding missing elements, and extending the sequences was at the basic and advanced levels. (3) The results of the study showed the reciprocal correlation between achievements in mathematics and cognitive function of primary school students. The two phases that caused difficulties for students were revealed: understanding the problem and carrying out the plan phase.

## 1. Introduction

Children develop considerably in mathematical knowledge and skills and cognitive ability during their time in primary school. Classical educational theory asserts that what they learn has an influence on how they think, and mathematics learning is supposed to develop reasoning and problem solving [1]. Problem solving means finding a way out of a difficulty or an obstacle and attaining the aim that was not immediately attainable. Problem solving is a specific achievement of intelligence [2]. Reading and performing mathematics are the two most fundamental skills taught during the early years of formal education [3]. Perhaps the basic source of trouble in problem solving is that students cannot actively observe, check, and regulate their cognitive process when solving a problem [4]. There has been much less research examining the cognitive underpinnings of mathematics, but it appears that nonphonological skills, such as visuospatial skills and analog representation of numbers [5,6], as well as phonological memory skills [7,8], may be important for learning and performing mathematics.

Regulation of cognition involves activities used to check and monitor learning. These consist of planning activities (predicting outcomes, setting time strategies, using different forms of indirect trial and error, etc.) before solving the problem, checking activities (monitoring, testing, revising, and resetting one’s strategies for learning) in the process of learning, and controlling outcomes (assessing the outcomes of strategic actions with the criteria of effectiveness and efficiency) [9]. Villeneuve et al. [10] used multigroup structural equation models to examine the relationships between direct and indirect cognitive abilities and mathematics problem solving across six grade-level groups using the Kaufman Assessment Battery for Children and the Kaufman Tests of Educational Achievement [11]. After testing, they found direct and indirect relationships with mathematics problem solving, whereas the learning efficiency and retrieval fluency constructs had only an indirect relationship with mathematics problem solving via math computation.

A vast amount of literature has shown that cognitive abilities account for substantial variance in academic achievement [12,13]. The relationship between cognitive factors and academic achievement has been of interest for numerous researchers. Achievement in mathematics relies on one’s ability to understand and solve complex tasks that have an inherent logic, thereby increasing cognitive demands in this particular domain of study [14]. General cognitive abilities, which have been consistently related to mathematics achievements [15], seem to play an additional role, and mathematics anxiety is expressed as a feeling of fear that many people experience when engaging in mathematical tasks [16]. The skills associated with problem solving are an essential part of the cognitive domains of international educational assessments. Namely, tests such as Trends in International Mathematics and Science Survey (TIMSS) or Programme of International Student Achievement (PISA) include problems, which demand students to apply mathematical concepts and use mathematical reasoning to justify and support their answers. Consequently, problem solving and mathematical reasoning have an undoubted importance when facing the assessments of education [17]. From the point of view of learning, problem solving promotes and enhances the development of multiple skills, such as examining, representing, and implementing. The National Council of Teachers of Mathematics puts problem solving as one of the basic abilities required to equip students with mathematics skills [17]. Problem solving is a process of finding a solution to achieve certain goals [18]. According to Polya, the steps of problem solving can be performed by understanding problems or solving problems, arranging plans, carrying out the plan, and looking back [19]. It was established that, in order to understand a problem, it means having to express the information about the problem well and not just answering questions [20]. The information is related to the things that are known about the matters that are asked. Students can be considered to have understood the problem if they were able to reveal the data that are known and the data requested related to the problem at hand. The ability to uncover data and provide data involves the higher-order thinking skills [21,22]. Several studies showed that difficulties in solving mathematical problems may occur at any phase during performance (i.e., planning–execution–evaluation [23]), with the phases of planning and evaluation commonly regarded as more problematic. In this sense, students commonly demonstrate difficulties in planning how to execute the problem solving, using inadequate or insufficient strategies and devoting their efforts to performing calculations [24].

Cowan et al. [25] found that learning mathematics improves general cognitive abilities and indicated that the relationship between general cognitive abilities and mathematics learning is reciprocal, at least between the ages of 7 and 9. In the mathematical domain, various authors [26,27] have suggested that mathematical reasoning is facilitated by an individual’s capacity to interrelate spatial images and verbal propositions. Various studies have shown that students with a strong ability to solve spatial problems achieve good results in science and mathematics [28,29]. Moreover, using a between-subjects comparison of children with versus without mathematical learning disabilities, Geary and colleagues [30,31] demonstrated the relationship between cognitive abilities—including short-term memory, long-term memory retrieval, number comprehension, and knowledge—and acquisition of numerical and arithmetic knowledge in first and second graders. In addition, using correlational methods in a longitudinal study [32], the evidence of individual differences was found: the way in which phonological processing abilities and mathematical computation skills are related in the period from the second to the fifth grade. Passolunghi et al. [33] found that mathematics achievement is predicted *not* by phonological and counting performance but by short-term memory and working memory—the latter in particular. Specifically, working memory span measured both in the first and in the second grades was associated with good mathematics performance in the second grade. Taking everything into consideration, the novelty of our study is that we tested cognitive abilities of primary school students examining different problem-solving phases [33] by means of our created and validated diagnostic cognitive ability test for primary school students. The tasks in the test follow the mathematics curriculum for grade 4. According to the problem-solving model [34], it includes four phases: (1) understanding the problem, (2) arranging plans, (3) carrying out the plan, and (4) looking back to confirm the answer [34].

The main aim of this study was to determine the effect of achievements in mathematics on the cognitive abilities of primary school students.

The objectives were as follows:(1)To validate the methodology of the diagnostic test of cognitive abilities for primary school students;(2)To reveal the relationships of achievements in mathematics and cognitive abilities of primary school students.


## 2. Materials and Methods

### 2.1. Participants

The students examined in this study were randomly selected from three state primary schools from various regions in Lithuania. The three selected schools follow the Lithuanian education system of primary, basic, and secondary education programs approved by the Lithuanian Minister of Education and Science in 2015.

The time and place of the study, with the consent of the parents, were agreed upon in advance with the school administration. This study was approved by the research ethics committee of Kaunas University of Technology, Institute of Social Science and Humanity (protocol no. V19-1253-03).

The participants selected from the 3 schools were 100 girls and 102 boys aged 9–10 years (fourth grade).

#### Measures

The diagnostic test of cognitive abilities (DTCA) was performed in the classroom, i.e., in a setting that was familiar to students. It was administered by the class teacher. Before the test, students were instructed briefly: the duration of the test—45 min, calculators were not allowed, worksheets could be used to carry out the calculations. The teacher would inform the students when 5 min was left to finish the test.

After the tests had been completed, the tests were collected and assessed by the teacher, and the results were sent to the examiner. The examiner evaluated students’ responses and investigated the mistakes made in the responses. The first phase included students’ unmarked answers. In some assignments, one of the answers was placed in an unplausible distractor. These selected responses were also assigned to the first phase. If the student’s chosen answer was a plausible distractor, the examiner checked the worksheets of the student’s solution to the problem and evaluated the mistakes made in the solutions. If there was no solution to the problem and the examiner could not attribute the incorrect answer to any problem-solving phase, they returned those worksheets to the teacher. After receiving the student’s mistakes from the examiner, the teacher used a think-aloud methodology to find out the mistakes and their causes, using Polya’s [34] problem-solving criteria and the think-aloud [35] methodology. Thus, it was possible to identify inappropriate operations, mistakes, or their causes in the students’ reasoning [35] and the phase of the problem solving in which the student made a mistake. The mistakes attributed by teachers to a particular problem-solving phase are not reflected in the results of this study. Only the data collected by the examiner were included in the statistical analysis to identify the problem-solving phases in which students made errors.

### 2.2. Instruments

#### 2.2.1. Rationale for the Diagnostic Test of Cognitive Abilities

The diagnostic test of cognitive abilities was created by the authors of the article for the assessment of primary school students’ cognitive abilities (further CA). The DTCA is based on Reuven Feuerstein’s theory of dynamic cognitive modality assessment [36] and the General Curriculum for Primary Education (approved by order no. ISAK-2433 of the Minister of Education and Science of the Republic of Lithuania, 26 August 2008 [37]). The test was designed for the 4-grade primary school students, covering the subject of mathematics. The DTCA is also based on the principles of individual assessment and specific assessment criteria. It is an objective and constructive way to determine the level of achievement of primary school students, allowing the planning of further teaching and learning in accordance with the student’s strengths and difficulties.

The purpose of the DTCA is to measure and assess the changes in primary school students’ knowledge and understanding, the application of knowledge and higher-order thinking skills. The tasks of the test were distributed according to the following cognitive functions:(1)systematic exploration (to asses this function, 3 tasks were included in the test): the function that is used to achieve systematic, non-impulsive, planned behavior in data collection. The learner creates a system (e.g., left to right, top to bottom) and uses it to complete the task sequentially (an example task is finding the differences between two pictures);(2)spatial orientation (2 tasks): the ability to perceive directions (in words or signs) and follow a given path (an example task is following a certain path indicated by arrows);(3)sequencing (3 tasks): the function used to define a rule for sequencing objects (an example task is setting a rule for the repetition of objects, numbers, or letters);(4)image recognition (2 tasks): the assessment of changes in visual objects after an action (an example task is indicating the order in which colored shapes are stacked);(5)recognizing and understanding relationships (1 task): the recognition of associations between elements by looking at their changes over time (an example task is arranging images in a logical sequence of events);(6)collecting and processing information (2 tasks): the ability to gather information accurately, clearly, and completely (an example task is recognizing the same objects after their positions are changed);(7)algorithm development (2 tasks): the ability to design/construct a logical rule tailored to a specific problem, regardless of the amount of data involved (an example task is figuring out how many times to cut a ribbon with scissors to obtain 4 pieces of ribbon);(8)data management (classification) (1 task): the classification of objects and events into groups or classes according to the defined criteria (an example task is sorting objects according to set or specified criteria);(9)construction of combinations (1 task): the construction of sets according to a given or created rule while recognizing the number of possibilities and variations in a combination (an example task is making possible combinations of specified objects).


Table 1 shows the structure of the diagnostic test of cognitive abilities based on the problem-solving model. According to the problem-solving model [36], it includes 4 phases: (1) understanding the problem, (2) arranging plans, (3) carrying out the plan, and (4) looking back to confirm the answer [34]. The diagnostic cognitive ability test is based on the problem-solving model and follows the mathematics curriculum for grade 4.

The DTCA scores tasks on a scale of 1 to 3. The number of points depends on the number of steps the student performs in the problem solution. For a problem with a score of 1, the student can obtain 0 points if the answer is not correct or 1 point if the answer is correct. For a task with a score of 2, the student may receive 0 points if the answer is incorrect and 2 points if the answer is correct. For a task with a score of 3, the student may obtain 0 points if the answer is incorrect, 1 or 2 points if the answer is partially correct, and 3 points if the answer is correct. After each task is scored, the test is marked on the total score. The test is corrected and graded by the teacher.

The tasks are also grouped according to levels of achievement and cognitive abilities. This DTCA matrix allows for the identification and assessment of levels of student achievement. The tasks in the test follow the mathematics curriculum for grade 4. Based on the characteristics of the DTCA and its scoring instructions, performance thresholds are set to ensure that primary school students are evaluated equally in terms of their cognitive abilities. DTCA defines an advanced level of achievement as a score between 22 and 29 points, basic between 15 and 21 points, satisfactory between 7 and 14 points, and low between 0 and 6 points. Cognitive ability clusters are used to identify levels of knowledge and understanding, application, and higher-order thinking skills.

The results of this assessment are used to determine how primary school students organize their learning process and how effectively they implement it. Based on the assessment of levels of achievement, the impact of learning methods on primary school students’ cognitive abilities is analyzed and interpreted.

An unsatisfactory level of achievement indicates that the student does not demonstrate the knowledge, understanding, and skills assessed in the cognitive ability group of the CA test.

A satisfactory level of achievement indicates that the student reproduces some knowledge but does not apply it to new situations and makes mistakes in standard mathematical procedures; has an insufficient understanding of mathematical concepts and symbols; can analyze individual details of a problem without associating them into a whole; has difficulty discerning patterns and relationships; recognizes familiar contexts and solves simple (often only one-step) problems; and chooses problem-solving strategies that are not always rational. The reasoning behind the decisions supports the conclusions, but these students do not notice errors in the decisions and therefore often draw incorrect conclusions. They do not provide any reasoning for their answers.

A basic level of achievement indicates that the student applies existing but not fully coherent knowledge to new and simple situations on the DTCA and demonstrates an understanding of and ability to perform standard mathematical procedures without making fundamental errors. The learner is able to read and understand the problem correctly but has a lack of precision and consistency in problem solving; thinks productively in common or familiar situations; can apply relationships between objects but only identifies the basic features, relationships, or patterns of objects; and solves problems correctly but does not interpret the final answer in the context of the original condition.

An advanced level of achievement indicates that the student has a good understanding of the terms of various problems, has learned and understands mathematical concepts, is able to perform standard mathematical procedures, and is able to solve mathematical and practical problems in different contexts. The learner demonstrates elements of creative thinking, is able to identify common and subordinate features of objects and their relationships, observes patterns, chooses the correct strategies to solve problems, and is able to test them. The student is able to draw detailed and accurate conclusions.

#### 2.2.2. Achievements in Mathematics

The methodology for assessing primary school students’ achievements is based on the General Curriculum for Primary Education (approved by the Order of the Minister of Education and Science of the Republic of Lithuania, no. ISAK-2433, 26 August 2008 [37] (Official Gazette of the Republic of Lithuania, 2008, No. 99-3848)) and the practice of Bambrick-Santoyo [38]. The mathematics performance assessment was designed for formative assessment of grade 4 primary school students. Mathematics achievement was measured systematically using short- and medium-term tests. Primary school students’ knowledge and skills were tested each week using a short test. The researcher prepared tasks to assess whether primary school students had achieved the learning objectives for the week. Except for rare cases, these objectives coincided with a theme of the yearly mathematics plan, as defined by the primary curriculum, which covers the following content areas: numbers and calculations, algebra, geometry, measures and measurement, and statistics. Primary school students were also taught how to solve math problems using strategies.

The first phase—understanding the problem (included in all 9 tasks). In this phase, primary school students are often stymied in their efforts to solve a problem simply because they do not understand it completely or understand just a part of the task. Teachers may encourage students by asking questions such as: Do you understand all the words used in stating the problem? What are you asked to find or show? Can you restate the problem in your own words? Can you think of a picture or diagram that might help you understand the problem? Is there enough information to enable you to find a solution? [19].

The second phase—arranging plans (in 9 tasks). In this phase, the skill of choosing an appropriate strategy is best learned by solving many problems. A partial list of strategies is included: guess and check, look for a pattern; make an orderly list, draw a picture, eliminate possibilities, solve a simpler problem, use symmetry; use a model, consider special cases, work backwards, use direct reasoning; use a formula, solve an equation, be ingenious [19].

The third phase—carrying out the plan (in 9 tasks). It is usually easier in this phase than in arranging the plan. While addressing the students, the teacher may stress the importance of care and patience as the most necessary skills in persisting with the plan that they have chosen [19].

The fourth phase—looking back to confirm the answer (in 9 tasks). Teachers used to teach their students to take the time to reflect and look back at what they have done, what worked, and what did not. Teachers stressed that doing this would enable the students to predict what strategy to use to solve future problems [19].

Short tests consisted of various numbers of tasks that met the following criteria: (1) the sum of the scores was 20 points, with 1 point per mathematical operation; (2) they were constructed by selecting tasks directly related to the mathematics subject content being taught; (3) they covered two areas of mathematical achievement, knowledge and understanding and communication, and the tasks were not at different levels of achievement; (4) they were at the same level of difficulty that the primary school students were studying that week; and (5) 30% of the test consisted of the tasks testing knowledge and understanding, while 70% tested knowledge application and skills.

Primary school students’ achievements were also measured by a mid-term test performed every 6 weeks, according to the following: (1) in the test, 90% of the content was directly related to the math topics covered during the 6 weeks, and 10% was a random selection of earlier topics that were still in the grade 4 curriculum. The tasks may have already been used in the short-term (weekly) tests. (2) The mid-term tests could have a varying number of tasks, but the sum of their scores was 40 points, with 1 point per mathematical operation. (3) The activities covered several areas of mathematical competence: knowledge and understanding, communication, mathematical reasoning, and problem-solving. The tasks were of different levels of achievement, covering the understanding of and the ability to apply knowledge in several areas of mathematics content, including more complex applications and problem solving. (4) The tasks were of varying difficulty, which should not match the level of difficulty at which they were taught, while maintaining the grade 4 level defined in the Primary Education Curriculum. (5) Among the tasks, 20–30% tested knowledge, 50% tested the ability to apply knowledge when solving mathematical tasks, and 20–30% tested problem-solving and advanced skills.

The achievement levels of mathematics, as was mentioned above, were based on Bambrick-Santoyo [38], the Primary Education Curriculum, and the Cambridge International Framework. The following levels of achievement were used to assess individual learner performance, define their learning strengths and weaknesses, and shape their subsequent learning process. The first level (1) score 80–100% means that the learner has a good understanding of the content, successfully achieves the objectives, and often exceeds expectations. The second level (2) score 60–80% means that the learner has a good understanding of the content of the curriculum and successfully achieves most of the learning objectives expected at this stage. The third level (3) score 40–60% means that the learner has a broad understanding of the content of the curriculum, achieves some of the learning objectives, and is working toward others, and would benefit from focusing more on some areas of the curriculum. The fourth level (4) score 0–40% means that the learner does not understand the content of the curriculum, does not achieve the learning objectives, and needs to pay more attention to certain areas of the curriculum.

### 2.3. Data Analysis

Graphic statistics are presented for all methodological factors as the mean ± SD (standard deviation). Spearman’s correlation coefficient was calculated to determine whether there was an association between cognitive functions. Statistical significance was defined as *p* ≤ 0.05 for all analyses. Analyses were carried out by utilizing SPSS 23 software (SPSS Inc., Chicago, IL, USA).

## 3. Results

The diagnostic test of cognitive abilities (DTCA) was analyzed in terms of the primary school students’ levels of achievement (satisfactory, basic, advanced) and cognitive functions. Diagnostic test of cognitive abilities (DTCA) has strong internal consistency (Cronbach’s Alpha was 0.728).

The analysis of data from the diagnostic test of cognitive skills and mathematics results of semester 1 led to the identification of four levels of learner achievement (unsatisfactory, satisfactory, basic, and advanced; see Figure 1). The results show that the majority of primary school students reached the basic level, with cognitive ability of 54% and mathematics achievement of 40%; all primary school students passed the unsatisfactory level with cognitive ability, and several primary school students reached mathematics achievement of 9%; knowledge was assessed at a satisfactory level, with cognitive ability of 26% and mathematics achievement of 14%; and the highest score in advanced level was cognitive ability of 20% and mathematics achievement of 37%.

The systematic exploration function was used to achieve systematic, non-impulsive, planned behavior when collecting data or checking information. It should be noted that almost all primary school students at the advanced level (score is 3.79 points out of maximum 4 points) were able to use this cognitive function, while just over half of the primary school students at the satisfactory level (score is 2.59 points out of maximum 4 points) were able to systematically gather information. There was a strong difference between satisfactory and advanced-level students in the ability to orient themselves in space and follow directions (scores are 0.94 and 2.71 points out of maximum 4 points, respectively). The distribution of scores for the item sequencing rule and finding missing items or extending the sequences was consistent with the achievement levels (satisfactory, 3.41 points; basic, 4.84 points; advanced, 5.57 points out of maximum 7 points), but the standard deviation exceeded 1 for all groups (1.73, 1.44, and 1.50, respectively). A small difference can be seen in the students’ ability to recognize an image after a certain change had been made. The difference between the results for satisfactory and advanced levels of achievement was 0.36 points. There was no difference in terms of the ability to identify and understand relationships when recognizing an association of elements in terms of change over time (score 1 out of 1, SD = 0). There was also very little difference in the collection and processing of information. Primary school students at the advanced level had a score of 2, SD = 0; those at the satisfactory level had a score of 1.76; and those at the basic level had a score of 1.97. Actually, the largest differences in performance were observed in multi-step tasks: designing an algorithm, classifying data and drawing conclusions, and constructing combinations. It should be noted that such tasks are difficult for primary school students at a satisfactory level of achievement; their scores were 0.71 points, 0.00 points, 0.35 points, respectively. The students at the basic level were able to complete these tasks correctly and obtain 1.79, 0.95, 1.42 points. The students at the advanced level were able to complete these tasks correctly, and their scores were 3.00, 1.93, 2.36 points out of maximum 4, 3, and 3 points, respectively (Table 2).

With regard to the relationships between cognitive functions, Spearman’s correlation analysis (Table 3) revealed the relationship between the following cognitive functions: systematic exploration and spatial orientation (Spearman q = 0.276, *p* = 0.022), systematic exploration and designing an algorithm (Spearman q = 0.351, *p* = 0.003), spatial orientation and data management (Spearman q = 0.274, *p* = 0.023), sequencing and combination construction (Spearman q = 0.275, *p* = 0.022), and sequencing and recognizing and understanding relationships (Spearman q = 0.243, *p* = 0.044). No statistically significant correlation was found between other cognitive functions.

To determine whether learning achievement in mathematics affects cognitive abilities, Spearman’s correlation analysis was performed (Figure 2). There is a direct moderate relationship between learning achievement and cognitive ability (Spearman q = 0.578, *p* = 0.000).

Examining the relationship between the cognitive function and mathematics achievement, Spearman’s correlation analysis (see Table 4) revealed existing relationships, and Pearson’s Chi-Square showed whether there was a statistically significant difference between the analyzed results: mathematics achievement and systematic exploration (Spearman q = 0.361, *p* = 0.002), mathematics achievement and spatial orientation (Spearman q = 0.424, *p* = 0.000; Chi-Square = 103.890, *p* = 0.044), mathematics achievement and sequencing (Spearman q = 0.279, *p* = 0.019; Chi-Square = 216.364, *p* = 0.003)), mathematics achievement and algorithm development (Spearman q = 0.284, *p* = 0.017), mathematics achievement and data management (classification) (Spearman q = 0.250, *p* = 0.037), and mathematics achievement and construction of combinations (classification) (Spearman q = 0.237), *p* = 0.049).

In Figure 3, the presented data show the number of students making mistakes (per cent) in different phases of problem solving, according to the levels of achievement of the cognitive test result. The results of the test show (see Figure 3), that students at the satisfactory level had the most difficulty in problem understanding; 21% of students at this level made mistakes in this phase. They also had difficulties in carrying out the plan phase (15%). The most difficult phase for students at the basic level was carrying out the plan phase (12%). They were slightly less likely to make mistakes in problem understanding (10%). Students at the highest level made similar mistakes in all phases (6–7%).

## 4. Discussion

Our first and foremost aim was to reveal the relationships between the achievements in mathematics and cognitive abilities of primary school grade 4 students. Secondly, our study aimed to validate the methodology of the diagnostic test of cognitive abilities for primary school students.

It is agreed that cognitive abilities are related to mathematics skills and problem solving. However, cognitive abilities are not the sole determinants of performance in academic and work settings [39]. Cowan et al. investigated the relations between mathematics and cognitive ability in primary school. They used a cross-lagged path analysis approach, which included measurements of mathematics and general cognitive ability at three ages (7, 9, and 10 years). The cross-lagged path between mathematics at 7 years old and general cognitive ability at 9 years old was stronger than the cross-lagged path between general cognitive ability at 7 years old and mathematics at 9 years old. Between the ages of 9 and 10, both the cross-lagged paths were of a similar strength and slightly weaker than the corresponding paths between the ages of 7 and 9 [25]. That served as an encouragement to develop our investigation of the primary school students’ cognitive functions and their correlation with the achievements in mathematics. The results of our study show that the diagnostic test of cognitive abilities created for that purpose has strong internal consistency with the wording of the statements that are clear for primary school students (Cronbach’s Alpha of 0.728). The correlation matrix of cognitive functions demonstrated the reliability of this scale. This suggests that this methodology can be offered for teachers to use for the assessment of the cognitive abilities of primary school students. When designing the diagnostic cognitive abilities test, we found that cognitive functions are highly relevant among primary school students, i.e., they are already encountered by students and used in a variety of tasks, not just in mathematical tasks. They include recognizing images, recognizing connections, gathering information, and drawing simple conclusions. It is also evident from the results of the study that such functions, which require creative, systematic thinking, data analysis and inference, the creation of new results from the available information, are more complicated and less common for many primary school students. Such complex cognitive functions make these types of tasks more difficult to solve and influence teaching and learning problem solving.

The results of the study showed that there is a reciprocal correlation between the achievements in mathematics and cognitive functions of primary school students. Lu et al. (2011) established that cognitive abilities, including working memory and intelligence, explained 17.8% and 36.4% of the variance in children’s mathematics scores. Domain-specific motivational constructs contributed only marginally to the prediction of school achievement in mathematics [40]. Cognitive skills predict academic performance, so schools that improve academic performance might also expect to improve cognitive skills [41]. Solving mathematical problems is a complex task that involves several distinct abilities that are essential in everyday life situations. Therefore, understanding the factors related to strong mathematical abilities is extremely important [42]. Earlier studies, however, did not examine whether mathematics abilities would increase over and above the cognitive abilities consistently linked to student performance in mathematics [43]. Among our results, the data from the diagnostic test of cognitive abilities showed that the majority of primary school students reached higher levels of achievement in a systematic inquiry (systematic, non-impulsive, planned behavior when collecting data or checking information). A difference was observed in the ability to navigate in space and follow directions among students. Primary school students’ performance in identifying the rule for the sequencing of elements, finding missing elements, and extending the sequences was at the basic and advanced levels. The previous study [44] showed that in the experimental group, the intervention had a positive impact on the achievements in mathematics. The primary school students’ learning achievements were positive in progressive mathematics. The study demonstrated higher achievements in mathematics among students with significant advances in their cognitive abilities of thinking and application [44].

Kampa et al. [45] found that large-scale assessments of both mathematical and verbal achievement cover general cognitive abilities and domain-specific achievement dimensions. It was established that cognitive ability involves the ability to reason, plan, and solve problems [46]. Similar results were found by Finn et al. [41] who reported substantial positive correlations between cognitive skills and achievement test scores, especially in mathematics. Iglesias-Sarmiento and Deano [47] studied the relationship between cognitive functioning and mathematical achievement of 114 students in the fourth, fifth, and sixth grades. Differences in cognitive performance were studied concurrently in three selected achievement groups: mathematical learning disability group, low achieving group, and typically achieving group. In this study, the performance of the cognitive processes, such as planning, attention, and simultaneous and successive processing, were assessed at the end of the academic course. Regression analysis revealed that simultaneous processing is a cognitive predictor of mathematical performance, although the phonological loop was also associated with higher achievement [46]. Comparing the TIMSS 2019 [48] mathematics results of grade 4 primary school students with previous years’ results, a slow but improving trend could be seen. The results of our study illustrate that to maintain or improve this, it is important to enable primary school students to understand the learning process and develop self-assessment skills. Our study highlights three phases of problem-solving ability in which students encountered difficulties. The results of the diagnostic test of cognitive abilities (DTCA) show that students at the satisfactory level had the most difficulty with understanding problems; 21% of students at this level made mistakes in this phase. They also had difficulties in carrying out the plan phase (15%). The most difficult phase for students at the basic level was the carrying out the plan phase (12%). They were slightly less likely to make mistakes in the understanding the problem phase (10%). Students at the highest level made similar mistakes in all phases (6–7%). Campos et al. [49] found that the mathematical domain, such as arithmetic word problems and measurement skills (e.g., length and area), seem to require executive cognitive functions. For this purpose, it is important to clarify and classify the cognitive functions involved in the learning process and to familiarize primary school students with this and give them the examples of where they can use such functions in certain activities, providing students with the learning environment in which they feel self-aware and that empowers them to find a path to success.

There are some limitations in our study. To begin with, only a teacher, not an examiner, could interview a student using think-aloud protocols. Then, the planning of the research of this nature required flexible collaboration between teachers and researchers. Data collection, assessment, and discussion required confidence between the research participants and the researchers. It is important to strengthen the teachers’ research competence so that teachers could feel equal research partners investigating their professional practice.

### Limitations

Only a teacher, not an examiner, could interview a student using think-aloud protocols. Planning research of this nature requires flexible collaboration between teachers and researchers. Confidence between the teacher and the researcher is important, strengthening the research competence. A math teacher would be able to level with cognitive knowledge, i.e., operate in problem-solving phases according to Polya [19].

## 5. Conclusions

The internal validity of the diagnostic test of cognitive abilities was supported by significant correlations between cognitive functions and achievement. This suggests that this methodology of the diagnostic cognitive ability test can be used to assess the cognitive abilities of primary school students.The diagnostic test of cognitive abilities showed that the majority of primary school students reached higher levels of achievement in a systematic inquiry (systematic, non-impulsive, planned behavior when collecting data or checking information). Differences were observed in the ability to navigate in space and follow directions for primary school students at the satisfactory or higher level. Primary school students’ performance in identifying the logic for the sequencing of elements, finding missing elements, and extending the sequences was at the basic and advanced levels.The two phases that caused difficulties for students, namely understanding the problem and carrying out the plan phases, were established. The results of the study showed the reciprocal correlation between achievements in mathematics and cognitive function of primary school students.

## Figures and Tables

**Figure 1 brainsci-12-00736-f001:**
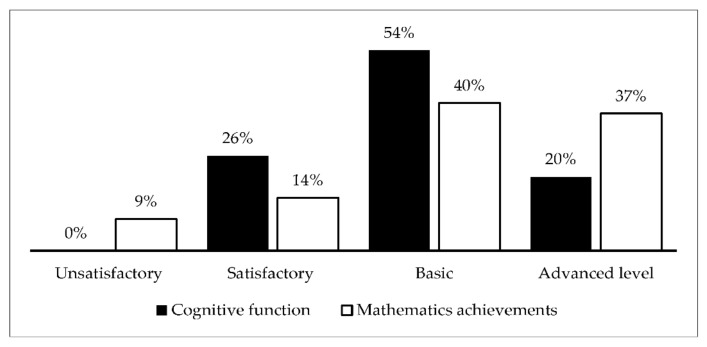
Achievements in mathematics and cognitive function of primary school students.

**Figure 2 brainsci-12-00736-f002:**
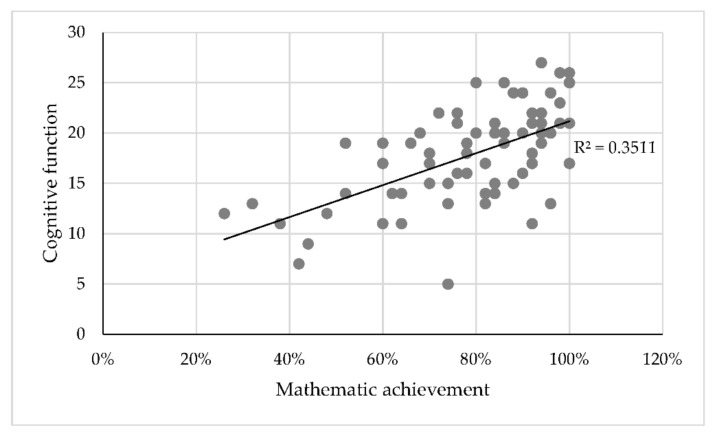
Relationship between mathematics achievement and cognitive function.

**Figure 3 brainsci-12-00736-f003:**
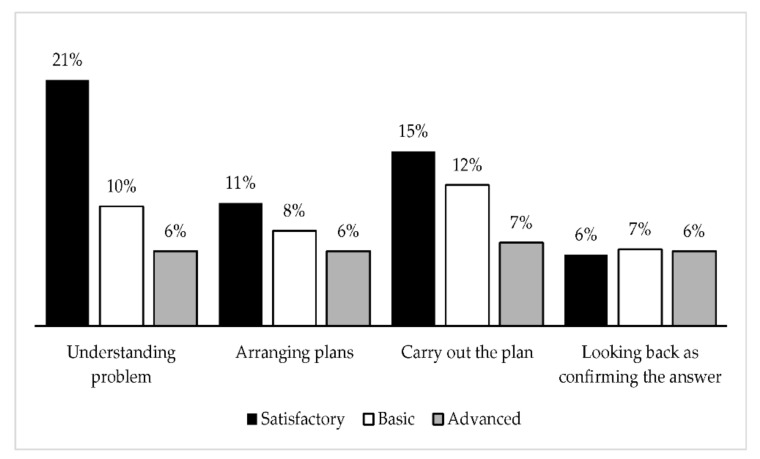
Primary schoolchildren problem-solving abilities phases.

**Table 1 brainsci-12-00736-t001:** Structure of diagnostic cognitive ability test.

Cognitive Function	Task	Problem-Solving Ability	Correct Answers
Systematic exploration	Kotryna colored all the squares on the table, which gave her 24. What did the table look like then? 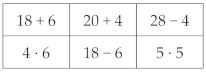	(a) is the correct answer; (c) and (d) show a mistake in phase 3, when performing the plan; (b) and (e) show a mistake in phase 1, misunderstanding the problem.	(a)
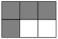	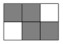	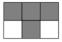	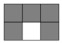	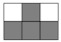
(a)	(b)	(c)	(d)	(e)
Collecting and processing information	The faster a swimmer finishes, the higher he or she stands on the podium. On which podium will the third-place swimmer stand? 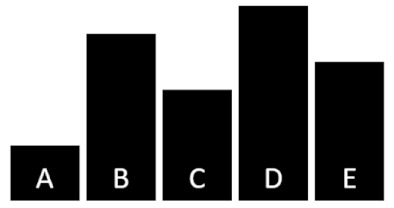	Answers (a), (b), and (d) indicate an error in phase 1. Answer (c) indicates an error in phase 2.	(e)
A	B	C	D	E
(a)	(b)	(c)	(d)	(e)
Image recognition	A book cover has two windows. When the book is opened, it looks like this: 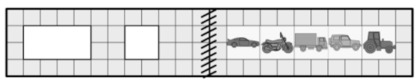 What images will you see through the windows when you close the book?	(a) indicates an error in phase 3, when carrying out the plan; the student understands that the page has to be covered, but the cover from the overlay shifts the image instead of flipping it.	(d)
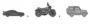	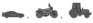	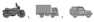
(a)	(b)	(c)
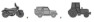		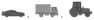
(d)		(e)
Recognizing and understanding connections	The flower grows every day. Which picture shows the flower on the second day?	Any answer other than E indicates an error in phase 3, when performing the plan.	(e)
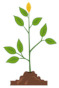	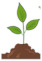	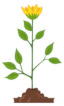	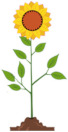	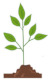
(a)	(b)	(c)	(d)	(e)
Orientation in space	Clouds must cover the suns. The arrows show how each cloud moves. Which suns will be covered by clouds?	The most common mistakes are taking one step instead of three, which indicates an error in phase 1; the other mistake is taking the wrong steps—an error in phase 3.	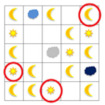
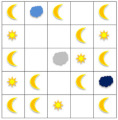	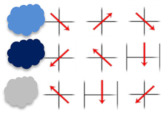
Orientation in space	Ina puts together a puzzle with 10 cards. She wants to put the crown out of 10 identical cards 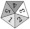 . The corresponding numbers on cards with a common edge must match. Four cards are already placed. What number will there be in the triangle marked with a question mark?	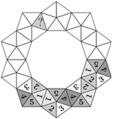	No answer indicates an error in phase 1. Mistake with shown calculation indicates an error in phase 3.	(d)
1	2	3	4	5
(a)	(b)	(c)	(d)	(e)
Sequences	Jurgis colors a drawing. He colors each flower petal red, yellow, or blue so that adjacent petals are different colors. He has already colored one petal blue. How many blue petals will there be in total when Jurgis has colored everything?	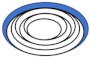	(b) indicates that the student’s plan was correct, a mistake was made in phase 3, when performing the plan, and the blue shape was not counted.	(c)
1	2	3	4	5
(a)	(b)	(c)	(d)	(e)
Creating an algorithm	In the factory, a bucket of blue paint is mixed every 7 min, and a bucket of red paint is mixed every 5 min. The packer stacks the buckets on the shelves as they come off the production line. The top shelf is filled first. Both production lines start work at the same time.	(a) and (b) answers indicate an error in phase 3 or 4. The student understood that the buckets of paint had to be mixed at different times but did not count the times correctly. (c) indicates that the student misunderstood the condition, so the error was in phase 1, i.e., in understanding the problem.	(d)
	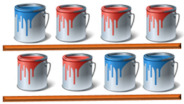
(a)	(b)
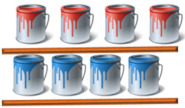	
(c)	(d)
Data processing (classification)	Nine participants took part in a turtles and rabbits running competition. Their scores were: 1, 2, 2, 3, 4, 5, 5, 6, 7. Unfortunately, the turtles were not so successful:No turtle beat any rabbit in points.One turtle finished in a tie with one rabbit.Two turtles were tied on points.How many rabbits and how many turtles took part in the competition?	If the sum given is 9, but it was not the right answer, an error was made in phase 3. If the answer is a number other than 9, the error was made in phase 1 or 2.	Turtles—6, rabbits—3.
Construction of combinations	Some children ordered ice cream shakes: 3 vanilla, 2 chocolate, and 1 strawberry. Three of them chose a cookie on top of their shakes, two chose whipped cream, and one chose sprinkles; then, there were no more identical shakes. Which shake did the children not have?(a)Chocolate with cookie(b)Vanilla with cookie(c)Strawberry with whipped cream(d)Chocolate with whipped cream(e)Vanilla with sprinkles	All incorrect answers indicate that the error was in phase 3, when performing the plan, because the condition automatically defines the process.	(c)

**Table 2 brainsci-12-00736-t002:** Assessment results for the diagnostic test of cognitive abilities by level of achievement (average (standard deviation)).

Achievement Level	Cognitive Function								
	Systematic Exploration	Spatial Orientation	Sequencing	Image Recognition	Recognizing and Understanding Relationships	Collecting and Processing Information	Designing an Algorithm	Data Management (Classification)	Construction of Combinations
Satisfactory	2.59 (1.37)	0.94 (0.83)	3.41 (1.73)	1.35 (0.70)	1.00 (0.00)	1.76 (0.44)	0.71 (0.99)	0.00 (0.00)	0.35 (1.00)
Basic	3.16 (0.92)	1.63 (1.08)	4.84 (1.44)	1.63 (0.49)	0.97 (0.16)	1.97 (0.16)	1.79 (1.45)	0.95 (1.41)	1.42 (1.52)
Advanced	3.79 (0.58)	2.71 (0.61)	5.57 (1.50)	1.71 (0.47)	1.00 (0.00)	2.00 (0.00)	3.00 (1.04)	1.93 (1.49)	2.36 (1.28)

**Table 3 brainsci-12-00736-t003:** Correlation matrix of cognitive functions tested.

		Systematic Exploration	Spatial Orientation	Sequencing	Image Recognition	Recognizing and Understanding Relationships	Collecting and Processing Information	Designing an Algorithm	Data Management (Classification)	Construction of Combinations
Systematic exploration	Correlation coefficient	1.000	**0.276 ***	−0.044	0.000	−0.168	0.045	**0.351 ****	0.042	−0.029
	Sig. (two-tailed)		**0.022**	0.717	0.999	0.167	0.711	**0.003**	0.733	0.812
Spatial orientation	Correlation coefficient		1.000	0.149	0.014	−0.132	0.123	0.200	**0.274 ***	0.206
	Sig. (two-tailed)			0.223	0.910	0.279	0.314	0.099	**0.023**	0.089
Sequencing	Correlation coefficient			1.000	0.088	**0.243 ***	0.072	0.006	0.025	**0.275 ***
	Sig. (two-tailed)				0.472	**0.044**	0.559	0.960	0.838	**0.022**
Image recognition	Correlation coefficient				1.000	−0.137	0.219	0.143	−0.033	0.201
	Sig. (two-tailed)					0.260	0.071	0.241	0.785	0.098
Recognizing and understanding relationships	Correlation coefficient					1.000	−0.048	−0.145	0.114	−0.191
	Sig. (two-tailed)						0.694	0.235	0.350	0.115
Collecting and processing information	Correlation coefficient						1.000	0.021	0.185	0.140
	Sig. (two-tailed)							0.863	0.128	0.251
Algorithm development	Correlation coefficient							1.000	−0.015	0.060
	Sig. (two-tailed)								0.901	0.625
Data management (classification)	Correlation coefficient								1.000	0.036
	Sig. (two-tailed)									0.770
Construction of combinations	Correlation coefficient									1.000
	Sig. (two-tailed)									

*, ** Correlation is significant at 0.05 and 0.01 level, respectively (two-tailed).

**Table 4 brainsci-12-00736-t004:** Spearman’s correlation between cognitive functions and mathematical achievement.

	Mathematics Achievement			
	Spearman Correlation	*p*	Pearson’s Chi-Square	*p*
Systematic exploration	**0.361 ****	**0.002**	129.416	0.078
Spatial orientation	**0.424 ****	**0.000**	**103.890 ***	**0.044**
Sequencing	**0.279 ***	**0.019**	**216.364 ***	**0.003**
Image recognition	0.186	0.123	66.413	0.120
Recognizing and understanding relationships	0.081	0.507	22.657	0.703
Collecting and processing information	0.233	0.052	64.932	0.147
Algorithm development	**0.284 ***	**0.017**	46.605	0.752
Data management (classification)	**0.250 ***	**0.037**	26.916	0.468
Construction of combinations	**0.237 ***	**0.049**	27.734	0.425

**—*p* < 0.01; *—*p* < 0.05.

## Data Availability

Not applicable.

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
