# Peer review of "The Impact of Achievements in Mathematics on Cognitive Ability in Primary School"

_brainsci, 2022, doi:10.3390/brainsci12060736_

Round 1
Reviewer 1 Report
The paper researches on a relevant problem dealing with cognitive abilities and their impact in mathematics. The consideration of cognitive abilities that are "in the border" of mathematics, such as algorithm development, data management and constructive abilities are particularly relevant (e.g. see P. Rosenbloom's concept of understanding vs shaping in his book "On computing...")
Though the research aim is promising and its research design seems correct (at least, up to the evidence available on the paper, not considering other issues regarding open data/science), the discussion and conclusions on the findings should be improved the discussion section is actually not discussing results, but exposing the related works. It should be improved to actually describe a discussion on the findings.
The writing of the Conclusion section should also be improved. It should not be a mere list of items.
Other aspects to improve:
- in the abstract, the conclusions paragraph refers to "this methodology". Which methodology? Please, put it in context within the abstract.
- page 9, line 286: "It is noted that such tasks are difficult for learners at a satisfactory level of achievement; their score was 10.6%" It seems that percentage is a mistake. The "level of achievement" should be a value in a 5-point scale instead of a percentage score.
- Figure 2 is not cited in the paragraph below it. It's confusing that the first column seems to represent a positive issue (correct answers), but the other three columns represent negative ones (students making mistakes). The explaining text is sometimes inconsistent with the figure legend. E.g. "13% of students made mistakes in solving tasks", but the text below the column is "understanding problem". Wrt the explanation of Fig. 2. (cf. "The results of the study highlight that the two phases that cause difficulties for students are understanding the problem and performing the plan."), why do "understanding" and "performing the plan" are the only ones which cause difficulties? The graph bar for "planning" is smaller than for "understanding the problem" (or problem-solving?) and "performing the plan".
Author Response
Firstly, the authors want to thank your contribution to our paper. We really appreciate you taking the time out to share your experience with us.
The paper researches on a relevant problem dealing with cognitive abilities and their impact in mathematics. The consideration of cognitive abilities that are "in the border" of mathematics, such as algorithm development, data management and constructive abilities are particularly relevant (e.g. see P. Rosenbloom's concept of understanding vs shaping in his book "On computing...")
Though the research aim is promising and its research design seems correct (at least, up to the evidence available on the paper, not considering other issues regarding open data/science), the discussion and conclusions on the findings should be improved the discussion section is actually not discussing results, but exposing the related works. It should be improved to actually describe a discussion on the findings.
Response:
It was corrected
The results of the study showed that the reciprocal correlation between achievements in mathematics and cognitive ability for primary schoolchildren. Lu et. al., 2011 established, that cognitive ability including working memory and intelligence explained 17.8% and 36.4% of the variance in children's mathematics scores. Domain-specific motivational constructs contributed only marginally to the prediction of school achievement for mathematics [38].
- . Lu, L.; Weber, S. H.; Spinath, F. M.; Shi, J. Predicting school achievement from cognitive and non-cognitive variables in a Chinese sample of elementary school children. 2011, 39(2-3), 130-140. https://doi.org/10.1016/j.intell.2011.02.002
The writing of the Conclusion section should also be improved. It should not be a mere list of items.
Response:
It was corrected
- The results of the study show relatively high internal and external validity and reliability of the diagnostic cognitive ability test. This suggests that this methodology of diagnostic cognitive ability test can be used to assess the cognitive abilities of primary school students.
- The results of the study showed that the reciprocal correlation between achievements in mathematics and cognitive ability for primary schoolchildren. Established highlight the two phases that cause difficulties for students: understanding the problem and performing the plan. It is notable that the majority of students were able to understand and plan a solution for the problem, but one-fifth of them made mistakes in planning, so it can be concluded that students do not have sufficiently developed, appropriate skills to perform a plan and reach the fourth test phase, confirming the answer.
Other aspects to improve:
- in the abstract, the conclusions paragraph refers to "this methodology". Which methodology? Please, put it in context within the abstract.
Response:
It was corrected
Abstract: Cognitive skills predict academic performance, so schools that improve academic performance might also improve cognitive skills. The purpose of this study was to determine the impact of achievements in mathematics on cognitive ability in primary school.
Methods: Participants, 100 girls and 102 boys aged 9–10 years (fourth grade), were selected from three schools. Diagnostic test of cognitive abilities (DTCA). DTCA based on the problem-solving model. It includes 4 phases: understanding problem, arranging plans, carry out the plan and looking back as confirming the answer. The tasks of the test were distributed according to cognitive function: systematic exploration, spatial orientation, sequencing, image recognition, recognizing and understanding relationships, collecting and processing information, algorithm development, data management (classification), and construction of combinations. Achievements in mathematics: The performance assessment was designed for formative assessment of grade 4 learners in mathematics.
Results: Established relationships between cognitive functions, Spearman's correlation analysis revealed relationships between the following cognitive functions: systematic exploration and spatial orientation (Spearman q = 0.276, p = 0.022), systematic exploration and designing an algorithm development (Spearman q = 0.351, p = 0.003), spatial orientation and data management (Spearman q = 0.274, p = 0.023), sequencing and combination construction (Spearman q = 0.275, p = 0.022) and sequencing and recognizing and understanding relationships (Spearman q = 0.243, p = 0.044).
Conclusions: (1) The results of the study show relatively high internal and external validity and reliability of the diagnostic cognitive ability test. This suggests that this methodology of DTCA can be used to assess the cognitive abilities of primary school students. (2) DTCA showed that the majority of learners reached higher levels of achievement in systematic inquiry (systematic, non-impulsive, planned behavior when collecting data or checking information). A difference is observed in the ability to navigate in space and follow directions for learners at a satisfactory or higher level. Learners' performance in identifying the rule for the sequencing of elements, finding missing elements, and extending the sequences was at the basic and advanced levels. The study shows that learners experienced difficulties with multi-step tasks that required them to develop an algorithm, classify the data, and construct combinations to draw conclusions. Such tasks were particularly challenging for learners at a satisfactory level of achievement. (3) The results of the study showed that the reciprocal correlation between achievements in mathematics and cognitive ability for primary schoolchildren.
- page 9, line 286: "It is noted that such tasks are difficult for learners at a satisfactory level of achievement; their score was 10.6%" It seems that percentage is a mistake. The "level of achievement" should be a value in a 5-point scale instead of a percentage score.
Response:
It was corrected
It is noted that such tasks are difficult for learners at a satisfactory level of achievement; their scores were 0.71 point, 0.00 point, 0.35 point, respectively. While of learners at the basic level were able to complete these tasks correctly and get 1.79, 0.95, 1.42 points. Learners at the advanced level were able to complete these tasks correctly and their scores were 3.00, 1.93, 2.36 points out of max 4, 3 and 3 points, respectively (Table 2).
- Figure 2 is not cited in the paragraph below it. It's confusing that the first column seems to represent a positive issue (correct answers), but the other three columns represent negative ones (students making mistakes). The explaining text is sometimes inconsistent with the figure legend. E.g. "13% of students made mistakes in solving tasks", but the text below the column is "understanding problem". Wrt the explanation of Fig. 2. (cf. "The results of the study highlight that the two phases that cause difficulties for students are understanding the problem and performing the plan."), why do "understanding" and "performing the plan" are the only ones which cause difficulties? The graph bar for "planning" is smaller than for "understanding the problem" (or problem-solving?) and "performing the plan".
Response:
It was corrected
The results of the test shows (Figure 3), that students at the satisfactory level had the most difficulty with understanding problems, 21% of students at this level have made mistakes in this phase. They also had difficulties in the carry out the plan phase (15%). The most difficult phase for students at the basic level was the carry out the plan phase (12%). They were slightly less likely to make mistakes in the understanding problem (10%). Students at the highest level have made similar mistakes in all phases (6-7%).

Reviewer 2 Report
Kliziene et al. investigate the relationship between achievements in mathematics and cognitive ability in 9-10 year-old children. The authors explore a number of cognitive functions, and report correlations between performance measures corresponding to different functions. The authors also explore differences in cognitive performance for individuals with different levels of achievement. In addition, the authors explore different phases of problem-solving ability. This study is ambitious, and the authors have certainly investigated a number of issues in this work. There are many pieces of data that could be very informative. However, I have reservations about the manuscript as it currently stands. At a high level, it is unclear from the manuscript that the work is very novel: we already know that cognition and mathematical performance must be linked in some way, from work that the authors review in the Introduction and Discussion, so the authors should clarify the contributions of this work in particular. In addition, from the title and various statements in the paper, the authors seem to claim that mathematical achievements are causing changes in cognitive ability, but I don’t see any results that suggest anything more than a correlation between these variables. In addition, despite reading the manuscript multiple times I found the methods and results difficult to follow, and I don’t see how the results support some of the claims the authors are making. My specific comments and suggestions are provided below:
Abstract
- Overall, the abstract is extremely long. From the Brain Sciences website it appears that the word limit on abstracts is 200 words, and the current abstract has over three times that number.
- We don’t know anything about the level scores at this point, so listing differences between the score levels for different cognitive functions in the results section is not very helpful. A more high-level description might be more useful.
- For the conclusions, it’s unclear how the results that were just listed pertain to internal/external validity/reliability (as I mention below, it’s unclear later on in the paper, too).
- Lines 53-55 or so: I assume you’re talking about satisfactory or higher levels of *math* performance, but this should be explicit
- It’s not clear to me how most of the conclusions relate to the central question of the article, i.e., the relation between math achievement and cognition.
Introduction
- I struggled a bit to understand the motivation for this work. Starting on line 100, the authors mention the Cowan et al. study that demonstrated a reciprocal relationship between mathematics learning and general cognitive ability between the ages of 7 and 9. The current study examines the same general question, but in slightly older children. It seems to me that the authors could better emphasize more novel questions that they are asking, like examining different problem-solving phases.
- Speaking of problem-solving phases, the authors don’t really mention this at all until the last sentence of the introduction. This should be expanded upon a great deal in the introduction. What do the authors mean by problem-solving ability phases? What are the hypothesized stages? Why are they important? What work has been done on this issue? Critically, later on, in the Methods, the authors briefly mention the 4 stages that they’re focusing on, and how this framework comes from a model that has been previously published. The authors should talk about this model in the Introduction – what the stages are, what data, if any, have supported the model, why they chose this model in particular, etc. It should also be more clear how the problem-solving phases relate to the larger question of impacts of learning mathematics on cognition.
- Minor: should line 87-88 be “...examine the direct and indirect relationships between cognitive ability and mathematics…”?
Materials and Methods
- I can’t tell if the CA diagnostic test was created by the authors for this study, or if it was an existing test that the authors utilized. If it’s novel, that should be emphasized more as a contribution of the study.
- The authors mention an example task for each cognitive function, but how many tasks were there for each function?
- How are the tests administered? Are they administered in person by an experimenter? By the classroom teachers?
- How are the tests scored? Are the points based on the number of tasks completed correctly? Again, is that coded by an experimenter who is in the room with the child, or by a teacher?
- I had a very difficult time understanding how the hypothesized problem-solving phases relate to the tasks in Table 1. The Table’s rightmost column makes claims about how specific incorrect answer choices map onto different phases, but it’s completely unclear to me how the authors came to these conclusions. For example, for the top row (systematic exploration), I don’t understand why answers C and D would indicate a mistake in performing the plan, whereas answers B and E would indicate misunderstanding the problem. As another example, why couldn’t an error in recognizing and understanding connections result from misunderstanding the problem (the authors conclude that any error would indicate an error in performing the plan)? In general, the mappings between errors and phases seem post hoc and not systematic.
- On a more minor note about Table 1, the correct answer choice should be noted for each of the tasks.
- Another minor Table 1 note – the problem-solving ability column mentions a gray shape for the “sequences” task, but I don’t see a gray shape in the image.
- The authors indicate on line 220 that the teacher prepared the math tasks, but if that’s the case how did the researchers ensure comparability across teachers/schools?
- Lines 250-256: the authors mentions 40-60%, 60-80%, and 80-100% levels of performance for math. It’s unclear where these numbers come from – is this the percentage of correctly answered questions? How do these levels relate to the short- and medium-term tests, and the midterm tests? In addition, do these levels map on to unsatisfactory/satisfactory/basic advanced levels of math achievement? Also, why are there not 20-40% or 0-20% levels?
Results
- The first sentence, on lines 263-264, mentions learners’ levels of achievement as satisfactory/basic/advanced. I assume this is referring to achievements in math? Do those correspond to the 40-60%, 60-80%, and 80-100% levels mentioned above? This should be much more clear. It’s also confusing because there are levels of achievement for both cognitive functions and math, and it’s often not explicit what the authors are referring to.
- Lines 264-265: what do the authors mean by systematic enquiry function?
- Speaking of the systematic enquiry function, the authors describe the learner creating a system and using it to complete the task sequentially (lines 266-267), but was this assessed through observation by an experimenter or by some other method?
- Much of the results section describes the distribution of cognitive scores as a function of achievement (presumably math achievement), which the authors take as evidence of an impact of math achievement on cognition, but there are no inferential statistics that could support these claims. The authors should do a regression or some other statistical analysis if they want to make claims about a relationship between math achievement and cognition.
- I am confused by the results pertaining to the distributions of cognitive functions (lines 263-288). I think that the numbers in this paragraph were referring to scores on the different cognitive tasks, but in some sentences it sounds like the authors are talking about numbers of learners (e.g., “ It should be noted that almost all learners at the advanced level (3.79 out of 4) were able to use this cognitive function, while just over half of the learners at the satisfactory level (2.59 out of 4) were able to systematically gather information”). This is confusing because the just-quoted sentences appear to be suggesting that there were 4 learners (participants), and 3.79 or 2.59 learners were able to do certain things, which doesn’t make sense, so I assume that those are average scores across all participants, but this needs to be much more clear.
- Lines 286-288: the authors mention percentages of learners at different levels, and reference Table 2, but Table 2 just has scores and what I assume are standard deviations, and not percentages of participants, so this is confusing.
- Speaking of Table 2, the authors should explicitly say in the caption what the numbers refer to (means and standard deviations of cognitive task scores?).
- Table 3: Why are there no correlations reported for recognizing and understanding relationships?
- Table 3: It looks like the last two rows are mislabeled, and should be data processing (classification), and construction of combinations.
- Figure 1: For achievements, I thought that satisfactory/basic/advanced corresponded to the 40-60%/60-80%/80-100% levels mentioned earlier, but those are only 3 levels, and here there are 4, so it’s confusing. Perhaps anything below 40% was “unsatisfactory”? Regardless, it needs to be much more clear where these levels and numbers are coming from.
- Figure 2: This figure is not referenced in the text. More importantly, I was even more confused by this figure than by Figure 1. From the description, it sounds like these are percentages of participants (e.g., “Among the students, 63% gave the correct answers to the research questions”). But this implies that some students always gave the correct answers for all tasks, or always made mistakes that indicated problems performing the plan, etc., which can’t be correct, so I assume these are percentages of all answer choices across all participants and tasks? This needs to be much more clear. I’m also unclear on how these results relate to the relationship between math and cognition. Perhaps the distribution of errors on cognitive tasks varied across levels of math achievement?
Discussion
- Lines 324-325: “The main aim of this study was to determine the impact of achievements in mathematics on cognitive ability in primary school.” As I mentioned above, this kind of language implies that the authors examined whether greater achievements in math causally impacts cognition, but I didn’t see any attempt to rule out other possibilities, including that differences in cognition caused differences in math performance, or that some other variable impacted both.
- Lines 325-327: The authors claim that the cognitive test has strong internal consistency, but I don’t see how the results support that claim (consistency isn’t mentioned in the results section).
- Similarly, on lines 327-328 the authors claim that correlations between performance for different cognitive functions demonstrate reliability, but again, it’s not clear what they are basing this on. There were a few significant correlations in Table 3, but the authors don’t discuss how the results relate to reliability (or validity).
Author Response
2 Review Report Form
Comments and Suggestions for Authors
Firstly, the authors want to thank your contribution to our paper. We really appreciate you taking the time out to share your experience with us.
Kliziene et al. investigate the relationship between achievements in mathematics and cognitive ability in 9-10 year-old children. The authors explore a number of cognitive functions, and report correlations between performance measures corresponding to different functions. The authors also explore differences in cognitive performance for individuals with different levels of achievement. In addition, the authors explore different phases of problem-solving ability. This study is ambitious, and the authors have certainly investigated a number of issues in this work. There are many pieces of data that could be very informative. However, I have reservations about the manuscript as it currently stands. At a high level, it is unclear from the manuscript that the work is very novel: we already know that cognition and mathematical performance must be linked in some way, from work that the authors review in the Introduction and Discussion, so the authors should clarify the contributions of this work in particular. In addition, from the title and various statements in the paper, the authors seem to claim that mathematical achievements are causing changes in cognitive ability, but I don’t see any results that suggest anything more than a correlation between these variables. In addition, despite reading the manuscript multiple times I found the methods and results difficult to follow, and I don’t see how the results support some of the claims the authors are making. My specific comments and suggestions are provided below:
Abstract
- Overall, the abstract is extremely long. From the Brain Sciences website it appears that the word limit on abstracts is 200 words, and the current abstract has over three times that number.
- We don’t know anything about the level scores at this point, so listing differences between the score levels for different cognitive functions in the results section is not very helpful. A more high-level description might be more useful.
- For the conclusions, it’s unclear how the results that were just listed pertain to internal/external validity/reliability (as I mention below, it’s unclear later on in the paper, too).
- Lines 53-55 or so: I assume you’re talking about satisfactory or higher levels of *math* performance, but this should be explicit
- It’s not clear to me how most of the conclusions relate to the central question of the article, i.e., the relation between math achievement and cognition.
Response:
It was corrected
Abstract: Cognitive skills predict academic performance, so schools that improve academic performance might also improve cognitive skills. The purpose of this study was to determine the impact of achievements in mathematics on cognitive ability in primary school.
Methods: Participants, 100 girls and 102 boys aged 9–10 years (fourth grade), were selected from three schools. Diagnostic test of cognitive abilities (DTCA). DTCA based on the problem-solving model. It includes 4 phases: understanding problem, arranging plans, carry out the plan and looking back as confirming the answer. The tasks of the test were distributed according to cognitive function: systematic exploration, spatial orientation, sequencing, image recognition, recognizing and understanding relationships, collecting and processing information, algorithm development, data management (classification), and construction of combinations. Achievements in mathematics: The performance assessment was designed for formative assessment of grade 4 learners in mathematics.
Results: Established relationships between cognitive functions, Spearman's correlation analysis revealed relationships between the following cognitive functions: systematic exploration and spatial orientation (Spearman q = 0.276, p = 0.022), systematic exploration and designing an algorithm development (Spearman q = 0.351, p = 0.003), spatial orientation and data management (Spearman q = 0.274, p = 0.023), sequencing and combination construction (Spearman q = 0.275, p = 0.022) and sequencing and recognizing and understanding relationships (Spearman q = 0.243, p = 0.044).
Conclusions: (1) The results of the study show relatively high internal and external validity and reliability of the diagnostic cognitive ability test. This suggests that this methodology of DTCA can be used to assess the cognitive abilities of primary school students. (2) DTCA showed that the majority of learners reached higher levels of achievement in systematic inquiry (systematic, non-impulsive, planned behavior when collecting data or checking information). A difference is observed in the ability to navigate in space and follow directions for learners at a satisfactory or higher level. Learners' performance in identifying the rule for the sequencing of elements, finding missing elements, and extending the sequences was at the basic and advanced levels. The study shows that learners experienced difficulties with multi-step tasks that required them to develop an algorithm, classify the data, and construct combinations to draw conclusions. Such tasks were particularly challenging for learners at a satisfactory level of achievement. (3) The results of the study showed that the reciprocal correlation between achievements in mathematics and cognitive ability for primary schoolchildren.
Introduction
- I struggled a bit to understand the motivation for this work. Starting on line 100, the authors mention the Cowan et al. study that demonstrated a reciprocal relationship between mathematics learning and general cognitive ability between the ages of 7 and 9. The current study examines the same general question, but in slightly older children. It seems to me that the authors could better emphasize more novel questions that they are asking, like examining different problem-solving phases.
- Speaking of problem-solving phases, the authors don’t really mention this at all until the last sentence of the introduction. This should be expanded upon a great deal in the introduction. What do the authors mean by problem-solving ability phases? What are the hypothesized stages? Why are they important? What work has been done on this issue? Critically, later on, in the Methods, the authors briefly mention the 4 stages that they’re focusing on, and how this framework comes from a model that has been previously published. The authors should talk about this model in the Introduction – what the stages are, what data, if any, have supported the model, why they chose this model in particular, etc. It should also be more clear how the problem-solving phases relate to the larger question of impacts of learning mathematics on cognition.
Response:
It was corrected
The skills associated to problem solving are an essential part of the cognitive domains of international educational assessments. Namely, tests like TIMSS or PISA include problems, which demand students to apply mathematical concepts and use mathematical reasoning to justify and support their answers. Consequently, problem solving and mathematical reasoning have an undoubtedly importance when facing educational assessments. From the point of view of learning, problem solving promotes and enhances the development of multiple skills, such as examining, representing, and, implementing. National Council of Teachers of Mathematicso puts problem-solving as one of the basic abilities required to get students to mathematics skills (National Council of Teachers of Mathematics, NCTM, 2000). Problem-solving is a process of finding a solution to achieve certain goals (Eichmann et al., 2019). According to Polya (1988), the steps of problem-solving can be done by understanding problems or solving problems, arranging plans, carry out the plan, and looking back. Estableshed, that to be able to understand a problem, it is mean, express information on the problem well and not just answer questions (Gulacar, Bowman, & Feakes, 2013). The information is related to things that are known about the matters that are asked. Students can be considered to have understood the problem if they were able to reveal the data that is known and the data requested related to the problem at hand. The ability to uncover data and provide data involves the higher-order thinking skills (Herranen & Aksela, 2019; Hu, Chiu, & Chiou, 2019).
Several studies showed that difficulties in solving mathematical problems may occur at any phase during performance (i.e., planning-execution-evaluation; Zimmerman, 2000), with the phases of planning and evaluation commonly regarded as more problematic. In this sense, students commonly demonstrate difficulties in planning how to execute the problem-solving, using inadequate or insufficient strategies and devoting their efforts to performing calculations (Garcia et al., 2019).
Cowan et al. [17] found that learning mathematics improves general cognitive abilities and indicated that the relationship between general cognitive ability and mathematics learning is reciprocal, at least between the ages of 7 and 9. In the mathematical domain, various authors [18, 19] have suggested that mathematical reasoning is facilitated by an individual’s capacity to interrelate spatial images and verbal propositions. Various studies have shown that students with a strong ability to solve spatial problems achieve good results in science and mathematics [20, 21]. Using a between-subjects comparison of children with versus without mathematical learning disabilities, Geary and colleagues (Geary, Hamson, & Hoard, 2000; Geary, Hoard, & Hamson, 1999) demonstrated the relationship between cognitive abilities—including short-term memory, long-term memory retrieval, number comprehension, and knowledge—and acquisition of numerical and arithmetic knowledge in first and second graders. Using correlational methods in a longitudinal study, Hecht, Torgesen, Wagner, and Rashotte (2001) found evidence of individual differences in how phonological processing ability and mathematical computation skills are related from second to fifth grades. Chiara Passolunghi et al. found that mathematics achievement is predicted not by phonological and counting performance, but by short tirm memory and working memory—the latter in particular. Specifically, working memory span measured both in first and in second grades was associated with good mathematics performance in second grade.
Our study novelty is created and validated the diagnostic cognitive ability test for primary school student, examining different problem-solving phases. The diagnostic cognitive ability test based on the problem-solving model. According by the problem-solving model by [24], it includes 4 phases: (1) understanding problem, (2) arranging plans, (3) carry out the plan, and (4) looking back as confirming the answer [24, Polya].
0 Hypothesis: Achievements in mathematics impact cognitive abilities of primary school students.
1 Hypothesis: Achievements in mathematics do not impact cognitive abilities of primary school students.
NCTM (2000). Principles and standards for school mathematics. Reston, VA: Author
Eichmann, B., Goldhammer, F., Greiff, S., Pucite, L., & Naumann, J. (2019). The role of planning in complex problem solving. Computers and Education, 128, 1–12.
Polya, G. (1988). How to solve it, a new aspect of mathematical method (2nd Ed.).
Gulacar, O., Bowman, C. R., & Feakes, D. a. (2013). Observational investigation of student 375 Yayuk & Husamah problem solving: The role and importance of habits. Science Education International, 24(2), 344–360
Herranen, J., & Aksela, M. (2019). Student-question-based inquiry in science education. Studies in Science Education, 55(1), 1–36
Hu, H. W., Chiu, C. H., & Chiou, G. F. (2019). Effects of question stem on pupils’ online questioning, science learning, and critical thinking. Journal of Educational Research, 112(4), 564–573.
- Zimmerman Attaining self-regulation. A social cognitive perspective M. Boekaerts, P.R. Pintrich, M. Zeidner (Eds.), Handbook of self-regulation, Academic, San Diego, CA (2000), pp. 13-39
Garcia T., Boom J., Kroesbergen E H., Nunez J C., Radriguez C. Planning, execution, and revision in mathematics problem solving: Does the order of the phases matter? Studies in Educational Evaluation, 2019, 61, 83-93
Geary, D. C., Hamson, C. O. and Hoard, M. K. 2000. Numerical and arithmetical cognition: A longitudinal study of process and concept deficits in children with learning disability. Journal of Experimental Child Psychology, 77: 236–263.
Geary, D. C., Hoard, M. K. and Hamson, C. O. 1999. Numerical and arithmetical cognition: Patterns of functions and deficits in children at risk for a mathematical disability. Journal of Experimental Child Psychology, 74: 213–239.
Hecht, S. A., Torgesen, J. K., Wagner, R. K. and Rashotte, C. A. 2001. The relations between phonological processing abilities and emerging individual differences in mathematical computation skills: A longitudinal study from second to fifth grades. Journal of Experimental Child Psychology, 79: 192–227.
Chiara Passolunghi. M., Mammarella, I. C., Altoè, G. 2008. Cognitive Abilities as Precursors of the Early Acquisition of Mathematical Skills During First Through Second Grades. Developmental Neuropsychology , 33 (3), 229-250.
- Minor: should line 87-88 be “...examine the direct and indirect relationships between cognitive ability and mathematics…”?
Response:
It was corrected
Villeneuve et al. [10] used multigroup structural equation models to examine the relationships between direct and indirect cognitive ability and mathematics problem-solving across six grade-level groups using the Kaufman Assessment Battery for Children and the Kaufman Tests of Educational Achievement [11]. After testing, they found direct and indirect relationships with mathematics problem-solving, whereas the learning efficiency and retrieval fluency constructs had only an indirect relationship with mathematics problem-solving via math computation.
Materials and Methods
- I can’t tell if the CA diagnostic test was created by the authors for this study, or if it was an existing test that the authors utilized. If it’s novel, that should be emphasized more as a contribution of the study.
- The authors mention an example task for each cognitive function, but how many tasks were there for each function?
Response:
It was corrected
The diagnostic test was created by the authors of the article for the assessment of primary school students’ cognitive abilities (further CA). The diagnostic CA test is based on Reuven Feuerstein's theory of dynamic cognitive modality assessment [22] and the General Curriculum for Primary Education (approved by order no. ISAK-2433 of the Minister of Education and Science of the Republic of Lithuania, 26 August 2008 [23]. The test is designed for learners in grade 4, covering the subject of mathematics. The diagnostic CA test, based on the principles of individual assessment and specific assessment criteria, is an objective and constructive way to determine level of achievement of learners, allowing for the planning of further teaching/learning in accordance with their strengths and difficulties.
The purpose of the CA test is to measure and assess changes in learners' knowledge and understanding, application of knowledge, and higher order thinking skills. The tasks in the test were distributed according to the following cognitive functions:
Systematic exploration (3 tasks): A function that is used to achieve systematic, non-impulsive, planned behavior in data collection. The learner creates a system (e.g., left to right, top to bottom) and uses it to complete the task sequentially. (An example task is finding the differences between two pictures).
Spatial orientation (2 tasks): The ability to perceive directions (in words or signs) and follow a given path. (An example task is following a certain path indicated by arrows.)
Sequencing (3 tasks): A function used to define a rule for sequencing objects. (An example task is setting a rule for the repetition of objects, numbers, or letters.)
Image recognition (2 tasks): The assessment of changes in visual objects after an action. (An example task is indicating the order in which colored shapes are stacked.)
Recognizing and understanding relationships (1 task): The recognition of associations between elements by looking at their changes over time. (An example task is arranging images in a logical sequence of events.)
Collecting and processing information (2 tasks): The ability to gather information accurately, clearly, and completely. (An example task is recognizing the same objects after their positions are changed.)
Algorithm development (2 tasks): The ability to design/construct a logical rule tailored to a specific problem, regardless of the amount of data involved. (An example task is figuring out how many times to cut a ribbon with scissors to get 4 pieces of ribbon.)
Data management (classification) (1 task): Classification of objects and events into groups or classes according to defined criteria. (An example task is sorting objects according to set or specified criteria.)
Construction of combinations (1 task): The construction of sets according to a given or created rule while recognizing the number of possibilities and variations in a combination. (An example task is making possible combinations of specified objects.)
Table 1 shows the diagnostic test of cognitive abilities based on the problem-solving model. According by the problem-solving model by [24], it includes 4 phases: (1) understanding problem, (2) arranging plans, (3) carry out the plan, and (4) looking back as confirming the answer [24, Polya].
- How are the tests administered? Are they administered in person by an experimenter? By the classroom teachers?
Response:
It was corrected
The test is performed in the classroom, i.e. in a setting that is familiar to students. It is administered by the class teacher. Before the test students are briefed on the instructions: the test is 45 minutes long, students cannot use calculators, and if necessary, they can use a worksheet to do the calculations. Teacher informs the students, when 5minutes are left to finish the test. The tests are then corrected by the teacher and the results are sent to the examiner.
- How are the tests scored? Are the points based on the number of tasks completed correctly? Again, is that coded by an experimenter who is in the room with the child, or by a teacher?
Response:
It was corrected
The test scores tasks on a scale of 1 to 3. The number of points depends on the number of steps the student performs in the problem solution. For a problem with a score of 1, the student can get 0 points if the answer is not correct or 1 point if the answer is correct. For a task with a score of 2, the student may receive 0 points if the answer is incorrect and 2 points if the answer is correct. For a task with a score of 3, the student may get 0 points if the answer is incorrect, 1 or 2 points if the answer is partially correct and 3 points if the answer is correct. After each task has been scored, the test is marked on the total score. The test is corrected and graded by the teacher
- I had a very difficult time understanding how the hypothesized problem-solving phases relate to the tasks in Table 1. The Table’s rightmost column makes claims about how specific incorrect answer choices map onto different phases, but it’s completely unclear to me how the authors came to these conclusions. For example, for the top row (systematic exploration), I don’t understand why answers C and D would indicate a mistake in performing the plan, whereas answers B and E would indicate misunderstanding the problem. As another example, why couldn’t an error in recognizing and understanding connections result from misunderstanding the problem (the authors conclude that any error would indicate an error in performing the plan)? In general, the mappings between errors and phases seem post hoc and not systematic.
Problem-solving phases nebuvo tirtos egzaminatoriui tiesiogiai vertinant kiekvieną eigos fazę. Pagal mokinių atliktus testus įvertinta, kurioje fazėje teoriškai galėjo būti padaryta klaida.
- On a more minor note about Table 1, the correct answer choice should be noted for each of the tasks.
Response:
It was corrected
- Another minor Table 1 note – the problem-solving ability column mentions a gray shape for the “sequences” task, but I don’t see a gray shape in the image.
Response:
It was corrected
Table 1. Structure of diagnostic cognitive ability test.
|
Cognitive function |
Task |
Problem-solving ability |
Correct answers |
||||||||||||||||
|
Systematic exploration |
Kotryna colored all the squares on the table, which gave her 24. What did the table look like then?
|
c) and d) show a mistake in phase 3, performing the plan. B and e) show a mistake in phase 1, misunderstanding problem |
a) |
||||||||||||||||
|
Collecting and processing information |
The faster a swimmer finishes, the higher he or she stands on the podium. On which podium will the third-place swimmer stand?
|
Any answer other than e) indicates an error in phase 3. |
e) |
||||||||||||||||
|
Image recognition |
A book cover has two windows. When the book is opened, it looks like this: What images will you see through the windows when you close the book?
|
a) indicates an error in phase 2, planning; closing the cover from the overlay shifts the image instead of flipping it. |
d) |
||||||||||||||||
|
Recognizing and understanding connections |
The flower grows every day. Which picture shows the flower on the second day?
|
Any answer other than e) indicates an error in phase 3, performing the plan. |
e) |
||||||||||||||||
|
Orientation in space |
Clouds must cover the suns. The arrows show how each cloud moves. Which suns will be covered by clouds?
|
The most common mistakes are taking one step instead of three or taking the wrong steps, an error in phase 3. In a rare error, the arrows indicate that students have not passed phase 1. |
|
Orientation in space |
|
|
d) |
|||||||||||||||
|
Sequences |
|
b) indicates that the student’s plan was correct, a mistake was made in phase 3, performing the plan, and blue shape was not counted. |
c) |
|||||||||||||||
|
Creating an algorithm |
In the factory, a bucket of blue paint is mixed every 7 minutes, and a bucket of red paint is mixed every 5 minutes. The packer stacks the buckets on the shelves as they come off the production line. The top shelf is filled first (from the left). Both production lines start work at the same time.
|
a) indicates an error in phase 3. The student understood that the lorries had to go differently and did not correctly count how differently. c) indicates that the student misunderstood the condition, so the error was in phase 1, understanding the problem. |
d) |
|||||||||||||||
|
Data processing (classification) |
Nine participants took part in a turtles and rabbits running competition. Their scores were: 1, 2, 2, 3, 4, 5, 5, 6, 7. Unfortunately, the turtles were not so successful: · No turtle beat any rabbit in points. · One turtle finished in a tie with one rabbit. · Two turtles were tied on points. How many rabbits and how many turtles took part in the competition? |
If the sum given is 9, then an error was made in phase 3, performing the plan. If the answer is a number other than nine, the error was made in phase 1 or 2. |
Turtles – 6, rabbits – 3. |
|||||||||||||||
|
Construction of combinations |
Some children ordered ice cream shakes: 3 vanilla, 2 chocolate, and 1 strawberry. Three of them chose a cookie on top of their shakes, two chose whipped cream, and one chose sprinkles, then there were no more identical shakes. Which shake did the children not have? a) Chocolate with cookie b) Vanilla with cookie c) Strawberry with whipped cream d) Chocolate with whipped cream e) Vanilla with sprinkles |
All incorrect answers indicate that the error was in phase 3, performing the plan, because the condition automatically defines the process. |
c) |
- The authors indicate on line 220 that the teacher prepared the math tasks, but if that’s the case how did the researchers ensure comparability across teachers/schools?
Response:
It was corrected
The researcher prepares tasks to assess whether learners have achieved the learning objectives for the week
- Lines 250-256: the authors mentions 40-60%, 60-80%, and 80-100% levels of performance for math. It’s unclear where these numbers come from – is this the percentage of correctly answered questions? How do these levels relate to the short- and medium-term tests, and the midterm tests? In addition, do these levels map on to unsatisfactory/satisfactory/basic advanced levels of math achievement? Also, why are there not 20-40% or 0-20% levels?
Response:
It was corrected
Achievement levels: based on Bambrick-Santoyo [25], the Primary Education Curriculum and the Cambridge International Framework, the following levels of achievement were used to assess individual learner performance, anticipate their learning strengths and weaknesses, and shape their subsequent learning process: (1) 80-100%: learner has a good understanding of the content, successfully achieves the objectives and often exceeds expectations; (2) 60-80%: learner has a good understanding of the content of the curriculum and successfully achieves most of the learning objectives expected at this stage; (3) 40-60%: learner has a broad understanding of the content of the curriculum, achieves some of the learning objectives, and is working toward others, and would benefit from focusing more on some areas of the curriculum, (4) 0-40%: the learner does not understand the content of the curriculum and does not achieve the learning objectives, and needs to pay more attention to certain areas of the curriculum.
Results
- The first sentence, on lines 263-264, mentions learners’ levels of achievement as satisfactory/basic/advanced. I assume this is referring to achievements in math? Do those correspond to the 40-60%, 60-80%, and 80-100% levels mentioned above? This should be much more clear. It’s also confusing because there are levels of achievement for both cognitive functions and math, and it’s often not explicit what the authors are referring to.
Response:
It was corrected
There are two levels of achievements: from cognitive test and mathematics achievement. Cognitive test has different scales and results are not counting in percent’s as in achievements in mathematics.
- Lines 264-265: what do the authors mean by systematic enquiry function?
Response:
It was corrected
The systematic exploration function was used to achieve systematic, non-impulsive, planned behavior when collecting data or checking information.
- Speaking of the systematic enquiry function, the authors describe the learner creating a system and using it to complete the task sequentially (lines 266-267), but was this assessed through observation by an experimenter or by some other method?
Response:
Correction error. The sentence was deleted in the text.
- Much of the results section describes the distribution of cognitive scores as a function of achievement (presumably math achievement), which the authors take as evidence of an impact of math achievement on cognition, but there are no inferential statistics that could support these claims. The authors should do a regression or some other statistical analysis if they want to make claims about a relationship between math achievement and cognition.
Response:
It was corrected
To determine whether learning mathematic achievement affects cognitive abilities, Spearman’s correlation analysis was performed (Figure XX). There is a direct moderate relationship between learning achievement and cognitive ability (Spearman q = 0.578, p = 0.000 (p <0.001)
Figure 2. Relationship between mathematic achievement and cognitive abilities
- I am confused by the results pertaining to the distributions of cognitive functions (lines 263-288). I think that the numbers in this paragraph were referring to scores on the different cognitive tasks, but in some sentences it sounds like the authors are talking about numbers of learners (e.g., “ It should be noted that almost all learners at the advanced level (3.79 out of 4) were able to use this cognitive function, while just over half of the learners at the satisfactory level (2.59 out of 4) were able to systematically gather information”). This is confusing because the just-quoted sentences appear to be suggesting that there were 4 learners (participants), and 3.79 or 2.59 learners were able to do certain things, which doesn’t make sense, so I assume that those are average scores across all participants, but this needs to be much more clear.
Response:
It was corrected
The diagnostic test of cognitive abilities was analyzed in terms of the learners' levels of achievement (satisfactory, basic, advanced) and cognitive functions. The systematic enquiry function was used to achieve systematic, non-impulsive, planned behavior when collecting data or checking information. The learner creates a system (e.g., left to right, top to bottom) and uses it to complete the task sequentially. It should be noted that almost all learners at the advanced level (score is 3.79 points out of max 4 points) were able to use this cognitive function, while just over half of the learners at the satisfactory level (score is 2.59 points out of max 4 points) were able to systematically gather information. There was a strong difference between satisfactory and advanced level learners in the ability to orient themselves in space and follow directions (scores are 0.94 and 2.71 points out of max 4 points, respectively). The distribution of scores for the item sequencing rule and finding missing items or extending the sequences was consistent with the achievement levels (satisfactory, 3.41 points; basic, 4.84 points; advanced, 5.57 points out of max 7 points), but the standard deviation exceeds 1 for all groups (1.73, 1.44, and 1.50 respectively).
- Lines 286-288: the authors mention percentages of learners at different levels, and reference Table 2, but Table 2 just has scores and what I assume are standard deviations, and not percentages of participants, so this is confusing.
Response:
It was corrected
Of note, the largest differences in performance were observed in multi-step tasks: designing an algorithm, classifying data and drawing conclusions, and constructing combinations. It is noted that such tasks are difficult for learners at a satisfactory level of achievement; their scores were 0.71 point, 0.00 point, 0.35 point, respectively. While of learners at the basic level were able to complete these tasks correctly and get 1.79, 0.95, 1.42 points. Learners at the advanced level were able to complete these tasks correctly and their scores were 3.00, 1.93, 2.36 points out of max 4, 3 and 3 points, respectively (Table 2).
- Speaking of Table 2, the authors should explicitly say in the caption what the numbers refer to (means and standard deviations of cognitive task scores?).
Response:
It was corrected
Table 2. Assessment results for the cognitive skills test by level of achievement (average (standart deviation))
- Table 3: Why are there no correlations reported for recognizing and understanding relationships?
Response:
It was corrected
Technical error
|
|
Systematic exploration |
Orientation in space |
Sequencing |
Image recognition |
Recognizing and understanding relationships |
Collecting and processing information |
Algorithm development |
Data processing (classification) |
Construction of combinations |
|
|
Systematic exploration |
Correlation coefficient |
1.000 |
0.276* |
-0.044 |
0.000 |
-0.168 |
0.045 |
0.351** |
0.042 |
-0.029 |
|
Sig. (2-tailed) |
|
0.022 |
0.717 |
0.999 |
0.167 |
0.711 |
0.003 |
0.733 |
0.812 |
|
|
Orientation in space |
Correlation coefficient |
|
1.000 |
0.149 |
0.014 |
-0.132 |
0.123 |
0.200 |
0.274* |
0.206 |
|
Sig. (2-tailed) |
|
|
0.223 |
0.910 |
0.279 |
0.314 |
0.099 |
0.023 |
0.089 |
|
|
Sequencing |
Correlation coefficient |
|
|
1.000 |
0.088 |
0.243* |
0.072 |
0.006 |
0.025 |
0.275* |
|
Sig. (2-tailed) |
|
|
|
0.472 |
0.044 |
0.559 |
0.960 |
0.838 |
0.022 |
|
|
Image recognition |
Correlation coefficient |
|
|
|
1.000 |
-0.137 |
0.219 |
0.143 |
-0.033 |
0.201 |
|
Sig. (2-tailed) |
|
|
|
|
0.260 |
0.071 |
0.241 |
0.785 |
0.098 |
|
|
Recognizing and understanding relationships |
Correlation coefficient |
|
|
|
|
1.000 |
-0.048 |
-0.145 |
0.114 |
-0.191 |
|
Sig. (2-tailed) |
|
|
|
|
|
0.694 |
0.235 |
0.350 |
0.115 |
|
|
Collecting and processing information |
Correlation coefficient |
|
|
|
|
|
1.000 |
0.021 |
0.185 |
0.140 |
|
Sig. (2-tailed) |
|
|
|
|
|
|
0.863 |
0.128 |
0.251 |
|
|
Algorithm development |
Correlation coefficient |
|
|
|
|
|
|
1.000 |
-0.015 |
0.060 |
|
Sig. (2-tailed) |
|
|
|
|
|
|
|
0.901 |
0.625 |
|
|
Data processing (classification) |
Correlation coefficient |
|
|
|
|
|
|
|
1.000 |
0.036 |
|
Sig. (2-tailed) |
|
|
|
|
|
|
|
|
0.770 |
|
|
Construction of combinations |
Correlation coefficient |
|
|
|
|
|
|
|
|
1.000 |
|
Sig. (2-tailed) |
|
|
|
|
|
|
|
|
|
|
|
*, ** Correlation is significant at 0.05 and 0.01 level, respectively (2-tailed). |
||||||||||
With regard to the relationships between cognitive functions, Spearman's correlation analysis (Table 3) revealed relationships between the following cognitive functions: systematic exploration and spatial orientation (Spearman q = 0.276, p = 0.022), systematic exploration and algorithm development (Spearman q = 0.351, p = 0.003), spatial orientation and data management (Spearman q = 0.274, p = 0.023), sequencing and combination construction (Spearman q = 0.275, p = 0.022), and sequencing and Recognizing and understanding relationships (Spearman q = 0.243, p = 0.044). No statistically significant correlation was found between other cognitive functions.
- Table 3: It looks like the last two rows are mislabeled, and should be data processing (classification), and construction of combinations.
Response:
It was corrected.
- Figure 1: For achievements, I thought that satisfactory/basic/advanced corresponded to the 40-60%/60-80%/80-100% levels mentioned earlier, but those are only 3 levels, and here there are 4, so it’s confusing. Perhaps anything below 40% was “unsatisfactory”? Regardless, it needs to be much more clear where these levels and numbers are coming from.
Response:
It was corrected
The results show that the majority of learners reached the basic level, with cognitive ability of 54% and mathematics achievement of 40%; all learners passed the unsatisfactory level, with cognitive ability and several learners reached mathematics achievement of 9%; knowledge was assessed at a satisfactory level, with cognitive ability of 24% and mathematics achievement of 14%; and the highest score in advanced level was cognitive ability of 20% and mathematics achievement of 37%.
- Figure 2: This figure is not referenced in the text. More importantly, I was even more confused by this figure than by Figure 1. From the description, it sounds like these are percentages of participants (e.g., “Among the students, 63% gave the correct answers to the research questions”). But this implies that some students always gave the correct answers for all tasks, or always made mistakes that indicated problems performing the plan, etc., which can’t be correct, so I assume these are percentages of all answer choices across all participants and tasks? This needs to be much more clear. I’m also unclear on how these results relate to the relationship between math and cognition. Perhaps the distribution of errors on cognitive tasks varied across levels of math achievement?
Response:
It was corrected
The results of the test shows (Figure 3), that students at the satisfactory level had the most difficulty with understanding problems, 21% of students at this level have made mistakes in this phase. They also had difficulties in the carry out the plan phase (15%). The most difficult phase for students at the basic level was the carry out the plan phase (12%). They were slightly less likely to make mistakes in the understanding problem (10%). Students at the highest level have made similar mistakes in all phases (6-7%).
Figure 3. Primary schoolchildren achievements in mathematics and problem-solving abilities phases
Discussion
- Lines 324-325: “The main aim of this study was to determine the impact of achievements in mathematics on cognitive ability in primary school.” As I mentioned above, this kind of language implies that the authors examined whether greater achievements in math causally impacts cognition, but I didn’t see any attempt to rule out other possibilities, including that differences in cognition caused differences in math performance, or that some other variable impacted both.
Response:
It was corrected.
The results of the study showed that the reciprocal correlation between achievements in mathematics and cognitive ability for primary schoolchildren.
- Lines 325-327: The authors claim that the cognitive test has strong internal consistency, but I don’t see how the results support that claim (consistency isn’t mentioned in the results section).
Response:
It was corrected.
Diagnostic test of cognitive abilities (DTCA) has strong internal consistency [Cronbachs Alpha] was 0.728)
- Similarly, on lines 327-328 the authors claim that correlations between performance for different cognitive functions demonstrate reliability, but again, it’s not clear what they are basing this on. There were a few significant correlations in Table 3, but the authors don’t discuss how the results relate to reliability (or validity).
systematic exploration and spatial orientation (Spearman q = 0.276, p = 0.022), systematic exploration and algorithm development (Spearman q = 0.351, p = 0.003), spatial orientation and data management (Spearman q = 0.274, p = 0.023), sequencing and combination construction (Spearman q = 0.275, p = 0.022), and sequencing and Recognizing and understanding relationships (Spearman q = 0.243, p = 0.044).

Round 2
Reviewer 1 Report
Revise English grammar of the added parts, particularly in the conclusions section.
Author Response
English grammar of the added parts, particularly in the conclusions section are done.

Reviewer 2 Report
The authors made some revisions to this manuscript in response to my first round of comments, and the manuscript is improved. However, there is still a long way to go. I still find the novelty of the study to be somewhat lacking. One idea for potentially improving this, as I explain below, is to correlate mathematics achievements with specific cognitive functions, or with specific problem-solving stages, in addition to overall cognitive scores. I don’t know if that would be possible – there may not be sufficient variability in the data – but perhaps it would show some interesting patterns. Otherwise, my biggest problem continues to be the problem-solving phases. I don’t understand how those were estimated from task performance. The only way that this is explained is by providing examples of how different mistakes indicate problems with different phases in Table 1, but as I mentioned in the first review, and bring up again below, it’s completely unclear how these were determined. Many of the examples of how particular errors correspond to different problem-solving phases seem arbitrary and post hoc. The problem-solving phases are a big part of the article, and the authors must do a better job of communicating how these were related to performance. Perhaps the authors could make a table showing specific criteria that were used to determine how errors corresponded to the phases.
Abstract
-
Lines 24-25: It should be specified which problem-solving model you’re using.
-
Results: You said at the beginning that the goal of the study is to predict cognitive ability based on mathematics performance, but here you’re talking about correlations between cognitive functions, and it’s unclear what these have to do with mathematics.
-
Conclusions (lines 38-39): It’s still unclear where the claim of high internal and external validity and reliability come from. To address reliability you added Cronbach’s alpha to the actual manuscript (not the abstract), but I’m still not sure where the internal and external validity claims are coming from. I’m assuming that these are based on the correlations between cognitive functions, but it’s not clear how the correlations you mention in the abstract relate to validity. As I mention below, it’s also unclear in the results and discussion sections of the man manuscript.
-
Conclusions (lines 42-43): “A difference is observed in the ability to navigate in space and follow directions for learners at a satisfactory or higher level.” I don’t know what this means. Are you saying that learners at a higher level were better able to navigate in space and follow directions than those at a satisfactory level? If so, a satisfactory or higher level of what, exactly? Overall cognitive performance? Or is this referring to levels of achievement in systematic inquiry that’s mentioned in the previous sentence? Or perhaps levels of math achievement? This kind of ambiguity is found throughout, e.g., lines 47-48 – “such tasks were particularly challenging for learners at a satisfactory level of achievement,” but achievement in what, exactly?
-
The Results and Conclusions sections in general seem to be focused on cognitive ability, but except for conclusion #3 (lines 48-50) I don’t know what these things have to do with mathematics.
Introduction
-
You should spell out what TIMSS and PISA stand for (line 89)
-
The section on problem-solving is a definite improvement over the first version of the manuscript.
-
It’s good that the authors made the novelty of their study explicit (lines 128-129), but the novelty still seems lacking to me. The authors mention they created and validated a diagnostic test, which is fine, but I’m wondering why it’s needed. As the authors point out, lots of studies have used cognitive tests, including in relation to mathematical ability, and so it’s unclear what this test in particular contributes. The authors mention that the test examines different problem-solving phases, which is good, but as the authors now point out in the introduction, several studies have already looked at problem-solving phases in relation to mathematics.
-
Lines 133-136 introduce two hypotheses – that achievements in mathematics impact cognitive abilities in primary school students, or not. However, as I mentioned in the first review, I don’t see how this study allows the authors to make a causal claim (i.e., that mathematics performance causes changes in cognition), which, to me, is inferred by the wording of the hypotheses. If the authors mean a more general hypothesis that the two variables are related, we know from a lot of studies that mathematics and cognition are related, as the authors discuss, so it seems like a foregone conclusion (of course they’re related).
Materials and Methods
-
Line 159: The authors should specify which test they’re talking about (presumably the DTCA).
-
Line 163: I think you mean “collected,” not corrected.
-
Table 1: I still don’t understand how the problem-solving stages were determined to relate to the cognitive task problems. As I mentioned in the first review, it still seems very arbitrary how different mistakes indicate failures of different problem-solving stages. For example, why does any incorrect choice on the flower problem (recognizing and understanding connections) indicate an error in performing the plan, and not an error in understanding the problem, arranging plans, or looking back to confirm the answer? It seems like an error could arise from any of those processes. I’m sure that the authors have some kind of system for mapping problem-solving phases onto their task, but it’s completely unclear to me what that system is, and given that the phases are an integral part of the paper this is a major issue.
-
A more minor note on Table 1: for the data processing example, the authors say “If the sum given is 9, then an error was made…” But the sum has to be 9 (the problem states that there were 9 participants, and the correct answer of 6 turtles and 3 rabbits sum to 9), so that’s clearly not an error.
-
Lines 211-218: I don’t understand the points system. The authors say the number of points depends on the number of steps performed by the student, but what does that mean? How do the authors know the number of steps? I’m assuming this goes back to the problem-solving stages, but again, I don’t know how those stages correspond to the tasks, so it’s unclear how the number of steps or stages was determined.
-
Lines 219-220: shouldn’t a Cronbach’s Alpha analysis go in the Results?
-
Lines 292-293: why should the level of difficulty of the midterm problems not match the level of difficulty at which they were taught? Do you mean to say that they SHOULD match?
-
Lines 300-308: the authors mention 0-40%, 40-60%, 60-80%, and 80-100% levels of performance for math. It’s unclear where these numbers come from – is this the percentage of correctly answered questions? How do these levels relate to the short- and medium-term tests, and the midterm tests? In addition, I’m guessing that these levels map on to unsatisfactory/satisfactory/basic/advanced levels of math achievement in Figure 1 but that should be explicit.
Results
-
Lines 315-316: “The diagnostic test of cognitive abilities was analyzed in terms of the learners' levels of achievement (satisfactory, basic, advanced) and cognitive functions.” Is “levels of achievement” referring to the overall cognitive levels of achievement (via DTCA), or levels of achievement in mathematics? This is important. If it’s referring to cognitive functions, which is the impression that I get, I’m not really sure what the point of this analysis is, because in that case all of these scores for the different cognitive functions went into determining students’ cognitive levels in the first place, right? In that case, the analysis seems to me like averaging three measures into one large measure, and then saying the three measures are correlated with the large measure – in that case, it wouldn’t be very interesting, because the large measure is a direct result of the three measures in the first place, so of course they’re correlated. I’m not saying that the analysis in the paper is useless, but it’s not clear what the point of the paragraph is and what knowledge it gives us, so the authors should give some motivation for why they’re breaking down these scores by level of achievement. If I’m wrong and levels of achievement refer to mathematics, then that would probably make more sense, but in any case the authors should really make clear exactly what they’re relating, here, and what the motivation for the analysis is.
-
It’s difficult to assess the number of points in that first Results paragraph and in Table 2, because the maximum number of points varies between the cognitive functions. I would suggest standardizing these scores, e.g. by making them proportions of correct responses (so that the maximum would be 1.0 for all measures) or by using some other method so that everything is on the same scale.
-
Lines 319-323: “It should be noted that almost all learners at the advanced level (score is 3.79 points out of max 4 points) were able to use this cognitive function, while just over half of the learners at the satisfactory level (score is 2.59 points out of max 4 points) were able to systematically gather information.” I’m confused by the systematically gathering information part – if the average score is 2.59 out of 4, and “just over half of the learners…were able to systematically gather information,” are you saying that a score of 2 out of 4 would indicate systematically gathering information?
-
From Table 2 it looks like there is no variability at all in recognizing and understanding relationships, if the average score was 1.0 for all levels with SD=0. I asked last time why there weren’t any correlations with that variable in Table 3, but now it makes sense why there weren’t, if there’s no variability. Now, I’m wondering how there ARE correlations in Table 3 if all participants got the same score and there’s no variability.
-
I don’t know how to interpret Table 3. The authors point out several significant correlations, but as far as I can tell the only discussion of this is in the Discussion section when the authors say “The correlation matrix of cognitive functions demonstrated the reliability of this scale.” I don’t know what the authors mean by this or how they came to that conclusion. The authors need more discussion on what the significant correlations mean, and what it means that other variables were not correlated. If the authors want to make a claim about reliability (or do they mean validity?), they need to explain their reasoning.
-
Line 360: What does it mean that “all learners passed the unsatisfactory level”? Does that mean that all learners were at least at that level? But if that was the lowest level of the scale isn’t that true by definition? Or is it saying that all learners were above unsatisfactory for the cognitive skills? In that case it’s confusing because the Figure shows that 1% were unsatisfactory.
-
Is the goal of lines 356-364 and Figure 1 just to show the distributions of levels, without making any particular claims about them? I wonder if it would make more sense to show that first, i.e., before showing differences in cognitive functions between levels.
-
I’m glad to see the correlation in Figure 2. Would it be possible to relate math achievement to the problem-solving phases in some way? I’m guessing there weren’t enough errors to allow that on a subject-level basis. Another possibility would be to relate math achievement to performance on the different cognitive functions (systematic exploration, etc.). In one way or another I think it would help make the study more novel if you could break cognition down in some way in relation to mathematics performance.
-
I don’t know how to interpret Figure 3 and lines 389-394. What does it mean that students in the satisfactory level had more trouble with understanding the problem, for example, whereas the advanced students had similar levels of mistakes across the phases? I’m also concerned that there aren’t any statistics here. Perhaps the authors could do a Chi-square test or something to see if there are significant differences between phases and/or levels of ability.
Discussion
-
As I mentioned for the Results, there are some results that need more explanation/discussion.
-
Last paragraph, lines 460-466: This needs to be updated since you’re now breaking down students into the levels of achievement.
-
Same (last) paragraph, lines 466-475: how does the study in [48] relate to your findings? Do you think that your study is in agreement with their study or are there important differences?
Conclusions
-
One of the main take-aways from this study seems to be the methodology of the novel cognitive test. This could be useful if the methodologies are made available to other researchers. Are you making the methodology available for other scientists to use in their own work?
Author Response
Review Report (Round 2)
Revise English grammar of the added parts, particularly in the conclusions section.
Reviewer 2
Review Report (Round 2)
The authors made some revisions to this manuscript in response to my first round of comments, and the manuscript is improved. However, there is still a long way to go. I still find the novelty of the study to be somewhat lacking. One idea for potentially improving this, as I explain below, is to correlate mathematics achievements with specific cognitive functions, or with specific problem-solving stages, in addition to overall cognitive scores. I don’t know if that would be possible – there may not be sufficient variability in the data – but perhaps it would show some interesting patterns. Otherwise, my biggest problem continues to be the problem-solving phases. I don’t understand how those were estimated from task performance. The only way that this is explained is by providing examples of how different mistakes indicate problems with different phases in Table 1, but as I mentioned in the first review, and bring up again below, it’s completely unclear how these were determined. Many of the examples of how particular errors correspond to different problem-solving phases seem arbitrary and post hoc. The problem-solving phases are a big part of the article, and the authors must do a better job of communicating how these were related to performance. Perhaps the authors could make a table showing specific criteria that were used to determine how errors corresponded to the phases.
Firstly, the authors want to thank your contribution to our paper. We really appreciate you taking the time out to share your experience with us.
Abstract
Lines 24-25: It should be specified which problem-solving model you’re using.
Response:
It was corrected
DTCA was created by the authors of the article for the assessment of primary school students’ cognitive abilities. The DTCA is based on Reuven Feuerstein's theory of dynamic cognitive modality assessment and the General Curriculum for Primary Education.
Results: You said at the beginning that the goal of the study is to predict cognitive ability based on mathematics performance, but here you’re talking about correlations between cognitive functions, and it’s unclear what these have to do with mathematics.
Response:
It was corrected
The results show that the majority of learners reached the basic level, with cognitive ability of 54% and mathematics achievement of 40%; all learners passed the unsatisfactory level, with cognitive ability and several learners reached mathematics achievement of 9%; knowledge was assessed at a satisfactory level, with cognitive ability of 26% and mathematics achievement of 14%; and the highest score in advanced level was cognitive ability of 20% and mathematics achievement of 37%.
Conclusions (lines 38-39): It’s still unclear where the claim of high internal and external validity and reliability come from. To address reliability you added Cronbach’s alpha to the actual manuscript (not the abstract), but I’m still not sure where the internal and external validity claims are coming from. I’m assuming that these are based on the correlations between cognitive functions, but it’s not clear how the correlations you mention in the abstract relate to validity. As I mention below, it’s also unclear in the results and discussion sections of the man manuscript.
Response:
It was corrected
The external validity of diagnostic test of cognitive abilities was supported by significant correlations between cognitive functions and measures
Conclusions (lines 42-43): “A difference is observed in the ability to navigate in space and follow directions for learners at a satisfactory or higher level.” I don’t know what this means. Are you saying that learners at a higher level were better able to navigate in space and follow directions than those at a satisfactory level? If so, a satisfactory or higher level of what, exactly? Overall cognitive performance? Or is this referring to levels of achievement in systematic inquiry that’s mentioned in the previous sentence? Or perhaps levels of math achievement? This kind of ambiguity is found throughout, e.g., lines 47-48 – “such tasks were particularly challenging for learners at a satisfactory level of achievement,” but achievement in what, exactly?
The Results and Conclusions sections in general seem to be focused on cognitive ability, but except for conclusion #3 (lines 48-50) I don’t know what these things have to do with mathematics.
Response:
It was corrected
Abstract: Cognitive skills predict academic performance, so schools that try to improve academic performance might also improve cognitive skills. The purpose of this study was to determine the effect of achievements in mathematics on cognitive ability in primary school. Methods: Participants: 100 girls and 102 boys aged 9–10 years (the fourth grade), were selected from three schools. Diagnostic test of cognitive abilities (DTCA) was created by the authors of the article for the assessment of primary school students’ cognitive abilities. The diagnostic cognitive ability test based on Reuven Feuerstein's theory of dynamic cognitive modality assessment, the problem-solving model and follow the mathematics curriculum for grade 4. The tasks of the test were distributed according to cognitive function: systematic exploration, spatial orientation, sequencing, image recognition, recognizing and understanding relationships, collecting and processing information, algorithm development, data management (classification), and construction of combinations. Achievements in mathematics: they were collected systematically using short- and medium-term mathematics tests and the levels of achievement were defined of grade 4 primary school students to assess individual learner performance, anticipate their learning strengths and weaknesses, and shape their subsequent learning process. Results: With regard to the relationships between cognitive functions and achievement level , Spearman's correlation analysis revealed the relationships between the following cognitive functions: systematic exploration and spatial orientation (Spearman q = 0.276, p = 0.022), systematic exploration and designing an algorithm development (Spearman q = 0.351, p = 0.003), spatial orientation and data management (Spearman q = 0.274, p = 0.023), sequencing and combination construction (Spearman q = 0.275, p = 0.022) and sequencing and recognizing and understanding relationships (Spearman q = 0.243, p = 0.044). Conclusions: (1) The internal validity of diagnostic test of cognitive abilities was supported by significant correlations between cognitive functions and mathematics achievement. This suggests that this methodology of diagnostic cognitive ability test can be used to assess the cognitive abilities of primary school students. (2) The diagnostic test of cognitive abilities showed that the majority of primary school students reached higher levels of achievement in systematic inquiry (systematic, non-impulsive, planned behavior when collecting data or checking information). A difference is observed in the ability of students to navigate in space and follow directions for primary school students at a satisfactory or higher level. Primary school students' performance in identifying the rule for the sequencing of elements, finding missing elements, and extending the sequences was at the basic and advanced levels. (3) The results of the study showed the reciprocal correlation between achievements in mathematics and cognitive function of primary school students. The two phases that cause difficulties for students were revealed: understanding the problem and carrying out the plan phase.
Introduction
You should spell out what TIMSS and PISA stand for (line 89)
Response:
It was corrected
Namely, tests like Trends in International Mathematics and Science Survey (TIMSS) or Programme of International Student Achievement (PISA) include problems, which demand students to apply mathematical concepts and use mathematical reasoning to justify and support their answers. Consequently, problem solving and mathematical reasoning have an undoubtedly importance when facing educational assessments.
The section on problem-solving is a definite improvement over the first version of the manuscript.
It’s good that the authors made the novelty of their study explicit (lines 128-129), but the novelty still seems lacking to me. The authors mention they created and validated a diagnostic test, which is fine, but I’m wondering why it’s needed. As the authors point out, lots of studies have used cognitive tests, including in relation to mathematical ability, and so it’s unclear what this test in particular contributes. The authors mention that the test examines different problem-solving phases, which is good, but as the authors now point out in the introduction, several studies have already looked at problem-solving phases in relation to mathematics.
Response:
It was corrected
Our study novelty is created and validated the diagnostic cognitive ability test for primary school student, examining different problem-solving phases [33]. The diagnostic cognitive ability test based on the problem-solving model. According by the problem-solving model by [34], it includes 4 phases: (1) understanding problem, (2) arranging plans, (3) carry out the plan, and (4) looking back as confirming the answer [34] and different cognitive functions: systematic exploration; collecting and processing information; image recognition; recognizing and understanding connections; orientation in space [35].
Lines 133-136 introduce two hypotheses – that achievements in mathematics impact cognitive abilities in primary school students, or not. However, as I mentioned in the first review, I don’t see how this study allows the authors to make a causal claim (i.e., that mathematics performance causes changes in cognition), which, to me, is inferred by the wording of the hypotheses. If the authors mean a more general hypothesis that the two variables are related, we know from a lot of studies that mathematics and cognition are related, as the authors discuss, so it seems like a foregone conclusion (of course they’re related).
Response:
We decided to highlight problematic issues at work
The objectives were as follows:
-
to validate the methodology of the diagnostic test of cognitive abilities for primary school students;
-
to reveal the relationships of achievements in mathematics and cognitive abilities of primary school students.
Materials and Methods
Line 159: The authors should specify which test they’re talking about (presumably the DTCA).
Line 163: I think you mean “collected,” not corrected.
Response:
It was corrected
2.1.1 Measures
2.1.1 Measures
The diagnostic test of cognitive abilities (DTCA) was performed in the classroom, i.e. in a setting that was familiar to students. It was administered by the class teacher. Before the test students were instructed briefly: the duration of the test – 45 minutes, calculators were not allowed to use, worksheets could be used to do the calculations. The teacher would inform the students when 5 minutes were left to finish the test.
After the test has been completed the tests were collected and assessed by the teacher and the results were sent to the examiner. The examiner evaluated students' responses and investigated the mistakes made in the responses. The first phase included students' unmarked answers. In some assignments one of the answers was placed in an unplausible distractor. These selected responses were also assigned to the first phase. If the student's chosen answer was a plausible distractor, the examiner checked the worksheets of the student's solution of the problem and evaluated the mistakes made in the solutions. If there was no solution of the problem and the examiner could not attribute the incorrect answer to any problem-solving phase, he returned those worksheets to the teacher. After receiving the student's mistakes from the examiner, the teacher used a think aloud methodology to find out the mistakes and their causes, using Polya' [34] problem-solving criteria and the think aloud [35] methodology. Thus, it was possible to identify inappropriate operations, mistakes or their causes in the students' reasoning [35], in which phase of the problem solving the student made a mistake. The mistake attributed by teachers to a particular problem-solving phase are not reflected in the results of this study. Only the data collected by the examiner were included in the statistical analysis to identify the problem-solving phases in which students made errors.
Table 1: I still don’t understand how the problem-solving stages were determined to relate to the cognitive task problems. As I mentioned in the first review, it still seems very arbitrary how different mistakes indicate failures of different problem-solving stages. For example, why does any incorrect choice on the flower problem (recognizing and understanding connections) indicate an error in performing the plan, and not an error in understanding the problem, arranging plans, or looking back to confirm the answer? It seems like an error could arise from any of those processes. I’m sure that the authors have some kind of system for mapping problem-solving phases onto their task, but it’s completely unclear to me what that system is, and given that the phases are an integral part of the paper this is a major issue.
Response:
It was corrected
The first phase – understanding the problem (included in all 9 tasks). In this phase primary school students are often stymied in their efforts to solve problems simply because they don’t understand it completely or understand just a part of the task. Teachers may encourage students asking questions such as: Do you understand all the words used in stating the problem? What are you asked to find or show? Can you restate the problem in your own words? Can you think of a picture or diagram that might help you understand the problem? Is there enough information to enable you to find a solution? [19].
The second phase – arranging plans (in 9 tasks). In this phase the skill of choosing an appropriate strategy is best learned by solving many problems. A partial list of strategies is included: guess and check, look for a pattern; make an orderly list, draw a picture, eliminate possibilities, solve a simpler problem, use symmetry; use a model, consider special cases, work backwards, use direct reasoning; use a formula, solve an equation, be ingenious [19].
The third phase – carrying out the plan (in 9 tasks). It is usually easier in this phase than in arranging the plan. While addressing the students the teacher may stress the importance of care and patience, as the most necessary skills persisting with the plan that they have chosen [19].
The forth phase – looking back as confirming the answer (in 9 tasks). Teachers used to teach their students to take the time to reflect and look back at what they have done, what worked, and what didn’t. Teachers stressed that doing this would enable the students to predict what strategy to use to solve future problems [19].
A more minor note on Table 1: for the data processing example, the authors say “If the sum given is 9, then an error was made…” But the sum has to be 9 (the problem states that there were 9 participants, and the correct answer of 6 turtles and 3 rabbits sum to 9), so that’s clearly not an error.
Response:
It was corrected
|
Cognitive |
Task |
Problem-solving |
Correct answers |
||||||||||||||||
|
Systematic |
Kotryna colored all the squares on the table, which gave her 24. What did the table look like then?
|
a) is the correct answer. c) and d) show a mistake in phase 3, performing the plan. b) and e) show a mistake in phase 1, misunderstanding the problem,. |
a) |
||||||||||||||||
|
Collecting and processing |
The faster a swimmer finishes, the higher he or she stands on the podium. On which podium will the third-place swimmer stand?
|
An answers a), b) and d) indicates an error in phase 1. Answer c) indicates an error in phase 2. |
e) |
||||||||||||||||
|
Image |
A book cover has two windows. When the book is opened, it looks like this:
What images will you see through the windows when you close the book?
|
a) indicates an error in phase 3, carring the plan: student understands, thar page has to be coverd, but the cover from the overlay shifts the image instead of flipping it. |
d) |
||||||||||||||||
|
Recognizing and |
The flower grows every day. Which picture shows the flower on the second day?
|
Any answer other than E indicates an error in phase 3, performing the plan. |
e) |
||||||||||||||||
|
Orientation in space |
Clouds must cover the suns. The arrows show how each cloud moves. Which suns will be covered by clouds?
|
The most common mistakes are taking one step instead of three, which indicate an error in phase 1. Other mistake is taking the wrong steps - an error in phase 3. |
|
|
Orientation in space |
|
No answer indicates an error in phase 1. Mistake with shown calculation indicates an error in phase 3. |
d) |
||||||||||||
|
Sequences |
|
b) indicates that the student’s plan was correct, a mistake was made in phase 3, performing the plan, and blue shape was not counted. |
c) |
||||||||||||
|
Creating an |
In the factory, a bucket of blue paint is mixed every 7 minutes, and a bucket of red paint is mixed every 5 minutes. The packer stacks the buckets on the shelves as they come off the production line. The top shelf is filled first. Both production lines start work at the same time.
|
a) and b) answers indicate an error in phase 3 or 4. The student understood that the buckets of paint had to be micxed go in differently time, but did not correctly count time correctly. c) indicates that the student misunderstood the condition, so the error was in phase 1, understanding the problem. |
d) |
||||||||||||
|
Data |
Nine participants took part in a turtles and rabbits running competition. Their scores were: 1, 2, 2, 3, 4, 5, 5, 6, 7. Unfortunately, the turtles were not so successful:
How many rabbits and how many turtles took part in the competition? |
If the sum given is 9, but it was not the right answer, an error was made in phase 3. If the answer is a number other than 9, the error was made in phase 1 or 2. |
Turtles – 6, rabbits – 3. |
||||||||||||
|
Construction of combinations |
Some children ordered ice cream shakes: 3 vanilla, 2 chocolate, and 1 strawberry. Three of them chose a cookie on top of their shakes, two chose whipped cream, and one chose sprinkles, then there were no more identical shakes. Which shake did the children not have?
|
All incorrect answers indicate that the error was in phase 3, performing the plan, because the condition automatically defines the process. |
c) |
Lines 211-218: I don’t understand the points system. The authors say the number of points depends on the number of steps performed by the student, but what does that mean? How do the authors know the number of steps? I’m assuming this goes back to the problem-solving stages, but again, I don’t know how those stages correspond to the tasks, so it’s unclear how the number of steps or stages was determined.
Response:
In math problems, 1 point is awarded as a standard for step 1 of the problem. If the task is three steps, such as calculating the sum, then the difference, and then the total product, then 3 points are awarded. There may be other evaluation systems.
Lines 219-220: shouldn’t a Cronbach’s Alpha analysis go in the Results?
Response:
It was corrected
Lines 292-293: why should the level of difficulty of the midterm problems not match the level of difficulty at which they were taught? Do you mean to say that they SHOULD match?
Response:
The tasks were of varying difficulty, which should not match the level of difficulty at which they were taught, while main training the grade 4 level defined in the Primary Education Curriculum
Lines 300-308: the authors mention 0-40%, 40-60%, 60-80%, and 80-100% levels of performance for math. It’s unclear where these numbers come from – is this the percentage of correctly answered questions? How do these levels relate to the short- and medium-term tests, and the midterm tests? In addition, I’m guessing that these levels map on to unsatisfactory/satisfactory/basic/advanced levels of math achievement in Figure 1 but that should be explicit.
Response:
Students' academic achievement is assessed by tests. The number of correct answers is converted to a percentage. The level of students ’mathematical achievement is determined by percentages, so, yes, it is the percentage of correct answers to questions.
Results
Lines 315-316: “The diagnostic test of cognitive abilities was analyzed in terms of the learners' levels of achievement (satisfactory, basic, advanced) and cognitive functions.” Is “levels of achievement” referring to the overall cognitive levels of achievement (via DTCA), or levels of achievement in mathematics? This is important. If it’s referring to cognitive functions, which is the impression that I get, I’m not really sure what the point of this analysis is, because in that case all of these scores for the different cognitive functions went into determining students’ cognitive levels in the first place, right? In that case, the analysis seems to me like averaging three measures into one large measure, and then saying the three measures are correlated with the large measure – in that case, it wouldn’t be very interesting, because the large measure is a direct result of the three measures in the first place, so of course they’re correlated. I’m not saying that the analysis in the paper is useless, but it’s not clear what the point of the paragraph is and what knowledge it gives us, so the authors should give some motivation for why they’re breaking down these scores by level of achievement. If I’m wrong and levels of achievement refer to mathematics, then that would probably make more sense, but in any case the authors should really make clear exactly what they’re relating, here, and what the motivation for the analysis is.
Response:
The meaning of this paragraph is to show that students solve DTCA tasks differently by dividing them according to cognitive ability achievement levels. Since the results of the cognitive test and the mathematical achievements of students are correlated, we believe that students who have acquired a certain level of mathematical achievement are able to solve certain CF tasks more successfully than those who have not. However, this is not self-evident, so this paragraph becomes a description of the results in the table.
It’s difficult to assess the number of points in that first Results paragraph and in Table 2, because the maximum number of points varies between the cognitive functions. I would suggest standardizing these scores, e.g. by making them proportions of correct responses (so that the maximum would be 1.0 for all measures) or by using some other method so that everything is on the same scale.
Response:
Mes nenorėtume visų kognityvinių funkcijų verčių Table 2 suvienodinti iki 1.0 balo ar 100 proc. Kadangi kiekvieną funkciją sudaro nevienodas uždavinių skaičius tai ir jos vertė yra skirtinga. Suvienodinus vertes neatsispindėtų funkcijų svoris, o ir procentinė ar 1 balo vertė nenusakytų kiekybinio rezultato.
Lines 319-323: “It should be noted that almost all learners at the advanced level (score is 3.79 points out of max 4 points) were able to use this cognitive function, while just over half of the learners at the satisfactory level (score is 2.59 points out of max 4 points) were able to systematically gather information.” I’m confused by the systematically gathering information part – if the average score is 2.59 out of 4, and “just over half of the learners…were able to systematically gather information,” are you saying that a score of 2 out of 4 would indicate systematically gathering information?
Response:
It was corrected.
The score of higher level students in this cognitive function is 3.79 points, the basic level is 3.16 points, and the satisfactory level is 2.59 points out of max 4 points.
From Table 2 it looks like there is no variability at all in recognizing and understanding relationships, if the average score was 1.0 for all levels with SD=0. I asked last time why there weren’t any correlations with that variable in Table 3, but now it makes sense why there weren’t, if there’s no variability. Now, I’m wondering how there ARE correlations in Table 3 if all participants got the same score and there’s no variability.
Response:
It was corrected
Technical error
Table 2. Assessment results for the cognitive function test by level of achievement (average (standard deviation))
|
Achievement level |
Cognitive function |
||||||||
|
Systematic exploration |
Spatial orientation |
Sequencing |
Image |
Recognizing and understanding relationships |
Collecting and processing information |
Designing an algorithm |
Data management (classification) |
Construction of combinations |
|
|
Satisfactory |
2.59 (1.37) |
0.94 (0.83) |
3.41 (1.73) |
1.35 (0.70) |
1.00 (0.00) |
1.76 (0.44) |
0.71 (0.99) |
0.00 (0.00) |
0.35 (1.00) |
|
Basic |
3.16 (0.92) |
1.63 (1.08) |
4.84 (1.44) |
1.63 (0.49) |
0.97 (0.16) |
1.97 (0.16) |
1.79 (1.45) |
0.95 (1.41) |
1.42 (1.52) |
|
Advanced |
3.79 (0.58) |
2.71 (0.61) |
5.57 (1.50) |
1.71 (0.47) |
1.00 (0.00) |
2.00 (0.00) |
3.00 (1.04) |
1.93 (1.49) |
2.36 (1.28) |
I don’t know how to interpret Table 3. The authors point out several significant correlations, but as far as I can tell the only discussion of this is in the Discussion section when the authors say “The correlation matrix of cognitive functions demonstrated the reliability of this scale.” I don’t know what the authors mean by this or how they came to that conclusion. The authors need more discussion on what the significant correlations mean, and what it means that other variables were not correlated. If the authors want to make a claim about reliability (or do they mean validity?), they need to explain their reasoning.
Response:
3 lentele norėta patikrinti ar egzistuoja ryšys tarp kognityvinių funkcijų, ką mes ir stebime, kad patikimas nors ir silpnas ryšys egzistuoja tarp keliatos kognityvinių funkcijų. Teoriniu požiūriu ryšių tarp kognityvinių funkcijų turėtų būti daugiau.
Line 360: What does it mean that “all learners passed the unsatisfactory level”? Does that mean that all learners were at least at that level? But if that was the lowest level of the scale isn’t that true by definition? Or is it saying that all learners were above unsatisfactory for the cognitive skills? In that case it’s confusing because the Figure shows that 1% were unsatisfactory.
Response:
Tai norima pasakyti, kad visi mokiniai surinko didesnę balų sumą ir perlipo nepatenkinamo lygio ribą. Grafike buvo įsivėlusi technonė klaida dėl 1% nepatenkinamo lygio.
Is the goal of lines 356-364 and Figure 1 just to show the distributions of levels, without making any particular claims about them? I wonder if it would make more sense to show that first, i.e., before showing differences in cognitive functions between levels.
Response:
Atsižvelgėm į jūsų gerą pastebėjimą dėl rezultatų pateikimo eiliškumo.
I’m glad to see the correlation in Figure 2. Would it be possible to relate math achievement to the problem-solving phases in some way? I’m guessing there weren’t enough errors to allow that on a subject-level basis. Another possibility would be to relate math achievement to performance on the different cognitive functions (systematic exploration, etc.). In one way or another I think it would help make the study more novel if you could break cognition down in some way in relation to mathematics performance.
Response:
Examining the relationship between cognitive function and Mathematics achievement, Spearman's correlation analysis (Table 4) revealed existing relationships, and Pearson Chi-Square showed whether there was a statistically significant difference between the analyzed results: mathematics achievement and systematic exploration (Spearman q = 0.361, p = 0.002), mathematics achievement and spatial orientation (Spearman q = 0.424, p = 0.000; Chi-Square = 103.890, p = 0.044), mathematics achievement and spatial orientation (Spearman q = 0.279, p = 0.019; Chi-Square = 216.364, p = 0.003) ), mathematics achievement and algorithm development (Spearman q = 0.284, p = 0.017), mathematics achievement and data management (classification) (Spearman q = 0.250, p = 0.037) and mathematics achievement and data management (classification) (Spearman q = 0.237) , p = 0.049).
Table 4. Spearman’s correlation between cognitive functions and mathematical achievement
|
|
Mathematics achievement |
|||
|
Spearman Correlation |
p |
Chi-Square |
p |
|
|
Systematic exploration |
0.361** |
0.002 |
129.416 |
0.078 |
|
Spatial orientation |
0.424** |
0.000 |
103.890* |
0.044 |
|
Sequencing |
0.279* |
0.019 |
216.364* |
0.003 |
|
Image recognition |
0.186 |
0.123 |
66.413 |
0.120 |
|
Recognizing and understanding relationships |
0.081 |
0.507 |
22.657 |
0.703 |
|
Collecting and processing information |
0.233 |
0.052 |
64.932 |
0.147 |
|
Algorithm development |
0.284* |
0.017 |
46.605 |
0.752 |
|
Data management (classification) |
0.250* |
0.037 |
26.916 |
0.468 |
|
Construction of combinations |
0.237* |
0.049 |
27.734 |
0.425 |
|
** – p < 0.01; * – p < 0.05 |
||||
I don’t know how to interpret Figure 3 and lines 389-394. What does it mean that students in the satisfactory level had more trouble with understanding the problem, for example, whereas the advanced students had similar levels of mistakes across the phases? I’m also concerned that there aren’t any statistics here. Perhaps the authors could do a Chi-square test or something to see if there are significant differences between phases and/or levels of ability.
Response:
Figure 3 the presented data show the amount of students making mistakes (per cent) in different phases of problem solving, according to the levels of achievement of the cognitive test result. Analizuojant mokinių klaidas, nustatyta, kad patenkinamame lygyje (pagal DTCA) 21 % mokinių atsakymų sudaro klaidos supratimo fazėje, 11 % mokinių atsakymų klaidos yra arranging plans fazėje, 15 % mokinių atsakymų – suklysta carry out the plan fazėje ir 6 % mokinių atsakymų yra klaidos looking back to confirming the answer fazėje. Pagrindiniame lygyje suklysta atitinkamose fazėse 10 %, 8 %, 12 % ir 7 % mokinių. Aukštesniajame lygyje klaidinų atsakymų yra 6 % mokinių pirmoje, antroje ir ketvirtoje fazėse, 7 % mokinių – trečioje, carry out the plan, fazėje.
Discussion
As I mentioned for the Results, there are some results that need more explanation/discussion.
Last paragraph, lines 460-466: This needs to be updated since you’re now breaking down students into the levels of achievement.
Same (last) paragraph, lines 466-475: how does the study in [48] relate to your findings? Do you think that your study is in agreement with their study or are there important differences?
Response:
It was corrected
Established, that an internal validity of diagnostic test of cognitive abilities was supported by significant correlations between cognitive functions and math, problem solving. Cognitive abilities are not the sole determinants of performance in academic and work settings [38]. Cowan et al., investigated the relations between mathematics and cognitive ability in primary school, they used a cross-lagged path analysis approach which includes measurements of mathematics and general cognitive ability at three ages (7, 9, and 10 years) [25]. The results of our study show that the diagnostic test of cognitive abilities has strong internal consistency, and the wording of the statements is clear for primary school students ([Cronbachs Alpha] was 0.728). The correlation matrix of cognitive functions demonstrated the reliability of this scale. This suggests that this methodology can be used to assess the cognitive abilities of primary school students. In the diagnostic cognitive abilities test, we found that cognitive functions that are more common among primary school students, i.e., they are already encountered by primary school students and used in a variety of tasks, not just mathematical tasks, stand out. These include recognizing images, recognizing connections, gathering information, and drawing simple conclusions. It is also evident from the results that such functions, which require creative, systematic thinking, data analysis and inference, and creating new results from the available information, are more complicated and less common for many primary school students, which makes these types of tasks more difficult to solve, which influences problem-solving.
The results of the study showed that the reciprocal correlation between achievements in mathematics and cognitive functions for primary schoolchildren. Lu et. al., 2011 established, that cognitive ability including working memory and intelligence explained 17.8% and 36.4% of the variance in children's mathematics scores. Domain-specific motivational constructs contributed only marginally to the prediction of school achievement for mathematics [39]. Cognitive skills predict academic performance, so schools that improve academic performance might also improve cognitive skills [40]. Solving mathematical problems is a complex task that involves several distinct abilities that are essential in everyday life situations. Therefore, understanding the factors related to strong mathematical abilities is extremely important [41]. Earlier studies, however, did not examine whether mathematics abilities would increase over and above cognitive abilities consistently linked to student performance in mathematics [42]. Cowan et al. [25] found a relationship between mathematics learning and general cognitive development between 7 and 9 years old. Researchers were estimate the cross-lagged effects of mathematics and general cognitive ability (measured as manifest rather than latent variables) between ages 7 and 9, and between ages 9 and 10, while allowing for the stability of both mathematics and general cognitive ability over time. It also allowed for residual covariances of mathematics and general cognitive ability within time point. The cross-lagged path between mathematics at 7 years old and general cognitive ability at 9 years old is stronger than the cross-lagged path between general cognitive ability at 7 years old and mathematics at years old ages. Between 9 and 10 ages, both the cross-lagged paths were of a similar strength, and slightly weaker than the corresponding paths between 7 and 9 ages. Our study found, that methodology of diagnostic test of cognitive abilities can be used to assess the cognitive abilities of primary school students [25]. Among our results, the data from the diagnostic test of cognitive abilities showed that the majority of primary school students reached higher levels of achievement in systematic inquiry (systematic, non-impulsive, planned behavior when collecting data or checking information). A difference is observed in the ability to navigate in space and follow directions for primary school students at a satisfactory or higher level. primary school students' performance in identifying the rule for the sequencing of elements, finding missing elements, and extending the sequences was at the basic and advanced levels. Previous study showed that in the experimental group, the intervention had a positive impact on access to mathematics, and primary school students learning achievements were positive in progressive mathematics. They demonstrated higher achievements in mathematics among school children, with significant advances in their cognitive abilities of thinking and application [43].
Kampa et al. [44] found that large-scale assessments of both mathematical and verbal achievement cover general cognitive abilities and domain-specific achievement dimensions. It was established that cognitive ability involves the ability to reason, plan and solve problems [45]. Similar results were found by Finn et al. [40], who reported substantial positive correlations between cognitive skills and achievement test scores, especially in mathematics. Iglesias-Sarmiento and Deano [46] studied the relationship between cognitive functioning and mathematical achievement in 114 students in fourth, fifth, and sixth grades. Differences in cognitive performance were studied concurrently in three selected achievement groups: mathematical learning disability group, low achieving group, and typical achieving group. For this study, performance in the cognitive processes of planning, attention, and simultaneous and successive processing was assessed at the end of the academic course. Regression analysis revealed that simultaneous processing is a cognitive predictor of mathematical performance, although the phonological loop was also associated with higher achievement [46]. Comparing the TIMSS 2019 [47] mathematics results for grade 4 primary school students with previous years' results, a slow but improving trend can be seen. To maintain or improve this, it is important to enable primary school students to understand the learning process and develop self-assessment skills. For this purpose, it is important to clarify and classify the thinking functions involved in the learning process and to familiarize primary school students with this process, and give them examples of where they can use which functions in certain activities. Providing a learning environment in which primary school students are self-aware of the learning process empowers them to be self-motivated to find a path to success. Experiencing success in the learning process is important for primary school primary school students.
Our study highlights three phases of problem-solving ability in which students encountered difficulties. The results of the diagnostic test of cognitive abilities (DTCA) shows, that students at the satisfactory level had the most difficulty with understanding problems, 21% of students at this level have made mistakes in this phase. They also had difficulties in the carry out the plan phase (15%). The most difficult phase for students at the basic level was the carry out the plan phase (12%). They were slightly less likely to make mistakes in the understanding problem (10%). Students at the highest level have made similar mistakes in all phases (6-7%). Campos et al. [48] found that the mathematical domain, such as arithmetic word problems and measurement skills (e.g., length and area), seem to require executive cognitive functions.
Conclusions
One of the main take-aways from this study seems to be the methodology of the novel cognitive test. This could be useful if the methodologies are made available to other researchers. Are you making the methodology available for other scientists to use in their own work?
The internal validity of diagnostic test of cognitive abilities was supported by significant correlations between cognitive functions and achievement. This suggests that this methodology of diagnostic cognitive ability test can be used to assess the cognitive abilities of primary school students.
